# Evolving Decomposed Plasticity Rules for Information-Bottlenecked Meta-Learning

**Fan Wang**[1,2]                                                                             *wang.fan@baidu.com*

**Hao Tian**[1]                                                                               *tianhao@baidu.com*

**Haoyi Xiong**[1]                                                                         *xionghaoyi@baidu.com*

**Hua Wu**[1]                                                                               *wu_hua@baidu.com*

**Jie Fu**[3]                                                                                 *fujie@baai.ac.cn*

**Yang Cao**[2]                                                                             *forrest@ustc.edu.cn*

**Yu Kang**[2]                                                                             *kangduyu@ustc.edu.cn*

**Haifeng Wang**[1]                                                                       *wanghaifeng@baidu.com*

[1]*Baidu Inc.*     [2]*University of Science and Technology of China*     [3]*Beijing Academy of Artificial Intelligence*

**Reviewed on OpenReview:** *https://openreview.net/forum?id=6qMKztPnOn*

## Abstract

Artificial neural networks (ANNs) are typically confined to accomplishing pre-defined tasks by learning a set of static parameters. In contrast, biological neural networks (BNNs) can adapt to various new tasks by continually updating the neural connections based on the inputs, which is aligned with the paradigm of learning effective learning rules in addition to static parameters, *e.g.*, meta-learning. Among various biologically inspired learning rules, Hebbian plasticity updates the neural network weights using local signals without the guide of an explicit target function, thus enabling an agent to learn automatically without human efforts. However, typical plastic ANNs using a large amount of meta-parameters violate the nature of the genomics bottleneck and potentially deteriorate the generalization capacity. This work proposes a new learning paradigm decomposing those connection-dependent plasticity rules into neuron-dependent rules thus accommodating $\Theta(n^2)$ learnable parameters with only $\Theta(n)$ meta-parameters. We also thoroughly study the effect of different neural modulation on plasticity. Our algorithms are tested in challenging random 2D maze environments, where the agents have to use their past experiences to shape the neural connections and improve their performances for the future. The results of our experiment validate the following: 1. Plasticity can be adopted to continually update a randomly initialized RNN to surpass pre-trained, more sophisticated recurrent models, especially when coming to long-term memorization. 2. Following the genomics bottleneck, the proposed decomposed plasticity can be comparable to or even more effective than canonical plasticity rules in some instances.

## 1 Introduction

Artificial Neural Networks (ANNs) with a vast number of parameters have achieved great success in various tasks (LeCun et al., 2015). Despite their capability of accomplishing pre-defined tasks, the generalizability to various new tasks at low costs is much questioned. Biological Neural Networks (BNNs) acquire new skills continually within their lifetime through neuronal plasticity (Hebb, 1949), a learning mechanism that shapes the neural connections based on local signals (pre-synaptic and post-synaptic neuronal states) only. In contrast, typical ANNs are trained once and for all and can hardly be applied to unseen tasks.

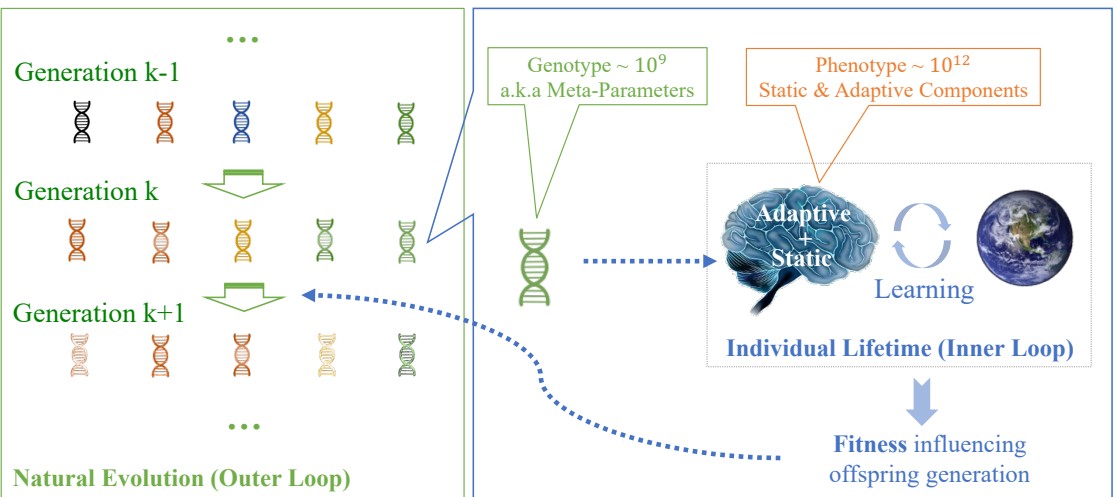

Figure 1: A simple illustration of the emergence of BNNs compared with meta-learning: The *genotype* evolved based on its fitness, which may be the result of individual development in its life cycle. The *phenotype* is initially dependent on the genotype but updated in its life cycle. We may more specifically divide it into the static components and adaptive components. The static components will not change within its life cycle, including the learning functionalities, initial neural architectures, and static neural connections. The adaptive components are continually updated with learning, including the neuron's internal states and plastic neural connections. The natural evolution and individual life cycle are similar to the nested learning loops of meta-learning, which include the *outer loop* and the *inner loop*. The genotype is similar to meta-parameters, and the phenotype corresponds to the parameters and hidden states in ANNs. Inspired by BNNs, the genotype that has a low capacity for information (e.g., $10^9$ base pairs in the DNAs of human beings) decides the learning mechanisms and the initialization of the phenotype. The phenotype has a massive capacity to memorize information (e.g. up to $10^{12}$ synapses in the human brain).

More recently, the emergence of BNNs has been used to inspire meta-learning (Zoph & Le, 2017; Finn et al., 2017). Instead of training a static neural network once and for all, meta-learning searches for learning rules, initialization settings, and model architectures that could be generalizable to various tasks. We therefore make an analogy between meta-learning and BNNs (Figure 1): The natural evolution and lifetime learning of BNNs correspond to the nested learning loops (outer loop and inner loop) in meta-learning; The *genotype* corresponds to the meta-parameters shaping the innate ability and the learning rules; The *phenotype* corresponds to the neural connections and hidden states that could be updated within the inner loop.

Although the full scope of the inner-loop learning mechanism of humans is not yet well known, investigations in this area have been used to interpret or even design ANN-based learning algorithms (Niv, 2009; Averbeck & Costa, 2017) for specific tasks. Those specific learning algorithms (including supervised learning, unsupervised learning, and reinforcement learning) can be applied directly to the inner loop in meta-learning (Finn et al., 2017). However, the majority of the previous works over-rely on the expert's efforts in objective function design, data cleaning, optimizer selection, etc., thus reducing the potential of automatically generalizing across tasks. Another class of meta-learning does not explicitly seek to understand the learning mechanism; Instead, they implicitly embed different learning algorithms in black-box mechanisms such as recursion (Duan et al., 2016) and Hebbian plasticity (Soltoggio et al., 2008; Najarro & Risi, 2020). Results have shown that these automated "black-box" learning mechanisms can be more sample efficient and less noise-sensitive than human-designed gradient-based learners. Unfortunately, they suffer from other defects: Model-based learners can be less effective when the life cycles are longer; Hebbian plasticity-based learners require many more meta-parameters than the neural connections, raising considerable challenges in meta-training.

Considering the analogy between meta-learning and the emergence of BNNs, most previous works fail to meet a significant hypothesis widely assumed in research on BNNs: *Genomic Bottleneck* (Zador, 2019; Pedersen & Risi, 2021; Koulakov et al., 2021). Compared to existing meta-learning algorithms that intensively rely on

pre-training many meta-parameters (genotype) (Yosinski et al., 2014; Finn et al., 2017) with less adaptive components (in the phenotype), BNNs actually acquire more information within the life cycle than those inherited from genotype (see Figure 1). In our opinion, the genomics bottleneck preserves relatively high learning potential while keeping the evolutionary process light. Previous research further finds that reducing meta-parameters also has a positive correlation with the generalizability of ANNs (Risi & Stanley, 2010; Pedersen & Risi, 2021).

Motivated by the aforementioned considerations, we propose a meta-learning framework with fewer meta-parameters and more adaptive components. The proposal is remarked with the following aspects:

- We revisit the canonical Hebbian plasticity rules (Hebb, 1949; Soltoggio et al., 2008) that employ 3 to 4 meta-parameters for each neural connection and propose *decomposed plasitcity*. Instead of assigning unique plasticity rules for each neural connection, we assume that the plasticity rules depend on the pre-synaptic neuron and post-synaptic neuron separately. As a result, we introduce a decomposition of the plasticity rules reducing the meta-parameters from $\Theta(n^2)$ to $\Theta(n)$ ($n$ is the number of neurons). For better generality and fewer meta-parameters, we further follow Najarro & Risi (2020) to force the plasticity rules to update neural connections from scratch, i.e., from random initialization, instead of searching for proper initialization of the connection weights.

- Following Miconi et al. (2018; 2019), we combine Hebbian plasticity with recursion-based learners in a meta-learning framework. But, in order to satisfy the genomic bottleneck, we go a step further by validating that plasticity rules can update the entire recursive neural network from scratch, including both the recurrent neural connections and the input neural connections. We also propose a method for visualizing plasticity-based learners, showing that plasticity is potentially more capable of long-term memorization than recurrence-based learners.

- Inspired by neural modulation in biological systems (Burrell & Sahley, 2001; Birmingham & Tauck, 2003), we investigate the details of neural modulation in plastic ANNs. We validate that even a single neural modulator can be crucial to plasticity. We also validate that the signals from which the modulator neurons integrate can significantly affect performance.

We select the tasks of 2D random maze navigation to validate our proposal, where the maze architectures, the agent origins, and the goals are randomly generated. The agents can only observe their surrounding locations and have no prior knowledge of the maze and the destination. Compared with the other benchmarks, it is able to generate endless new distinct tasks, thus effectively testing the agents' generalizability. The agents must preserve both short-term and long-term memory to localize themselves while exploiting shorter paths. Following the genomics bottleneck, we found that the decomposed plasticity yields comparable or even better performance than canonical plasticity rules while requiring fewer outer-loop learning steps. We also validate that plasticity can be a better long-term memorization mechanism than recurrence. For instance, our plastic RNNs surpass Meta-LSTM with over 20K meta-parameters in very challenging tasks by using only 1.3K meta-parameters.

## 2 Related Works

### 2.1 Deep Meta-Learning

In meta-learning, an agent gains experience in adapting to a distribution of tasks with nested learning loops: The outer learning loop optimizes the meta-parameters that may involve initializing settings (Finn et al., 2017; Song et al., 2019), learning rules (Li & Malik, 2016; Oh et al., 2020; Najarro & Risi, 2020; Pedersen & Risi, 2021), and model architectures (Zoph & Le, 2017; Liu et al., 2018; Real et al., 2019); The inner learning loops adapt the model to specific tasks by utilizing the meta-parameters. Based on the type of inner-loop learners, these methods can be roughly classified into *gradient-based* (Finn et al., 2017; Song et al., 2019), *model-based* (Santoro et al., 2016; Duan et al., 2016; Mishra et al., 2018; Wang et al., 2018), and *metric-based* (Koch et al., 2015) methods (Huisman et al., 2021). In addition, the *Plasticity-based* (Soltoggio et al., 2008; 2018; Najarro & Risi, 2020) methods update the connection weights of neural networks in the inner loop, but

not through gradients. A key advantage of plasticity and model-based learning is the capability of *learning without human-designed objectives and optimizers.* Our work combines both model-based and plasticity-based meta-learning.

## 2.2 Model-based Meta-Learning

Models with memories (including recurrence and self-attention) are capable of adapting to various tasks by continually updating their memory through forwarding (Hochreiter & Schmidhuber, 1997). Those models are found to be effective in automatically discovering supervised learning rules (Santoro et al., 2016), even complex reinforcement learning rules (Duan et al., 2016; Mishra et al., 2018). Similar capabilities are found in large-scale language models (Brown et al., 2020). Model-based learners own potential of unifying all different learning paradigms (supervised learning, unsupervised learning, reinforcement learning) within one paradigm. Still, the limitations of these learners become evident when life cycles get long. A reasonable guess is that the limited memory space restricted the learning potential since the adaptive components are typically much sparser than the static components (for the recurrent models, the adaptive components are the hidden states, which is in the order of $\Theta(n)$; The static components are the neural connections, which is $\Theta(n^2)$, with $n$ being the number of hidden units). In contrast, learning paradigms that update the neural connections have higher learning potential and better asymptotic performances.

## 2.3 Plastic Artificial Neural Networks

The synaptic plasticity of BNNs is found to depend on the pre-synaptic and post-synaptic neuronal states, which is initially raised by Hebb's rule (Hebb, 1949), known as "neurons that fire together wire together". Hebb's rule allows neural connections to be updated using only local signals, including the pre-synaptic and post-synaptic neuronal states. For ANNs, those rules are found ineffective without proper modulation and meta-parameters. For instance, in the $\alpha ABCD$ plasticity rule (Soltoggio et al., 2008), given the pre-synaptic neuron state $x_t$ and post-synaptic neuron state $y_t$, the neural connection weight $w_t$ is updated by

$$w_{t+1} = w_t + m_t[A \cdot x_t y_t + B \cdot x_t + C \cdot y_t + D], \tag{1}$$

where $A, B, C, D$ are meta-parameters depending on connections, and $m_t$ is the modulatory signal. In biological systems, one type of the most important modulatory neurons is *dopamine neurons*, which could integrate signals from the diverse areas of the nervous system (Watabe-Uchida et al., 2012) and affect the plasticity of certain neurons (Burrell & Sahley, 2001; Birmingham & Tauck, 2003). The effect of neural modulation on plastic ANNs is validated by Soltoggio et al. (2008). Other works uses simpler feedback signal (Frémaux & Gerstner, 2016), or trainable constants (Pedersen & Risi, 2021) as the modulation. The retroactive characteristic of dopamine neurons (Brzosko et al., 2015) also inspires the retroactive neuromodulated plasticity (Miconi et al., 2019), denoted by

$$\begin{aligned} w_{t+1} &= w_t + m_t e_{t+1}, \\ e_{t+1} &= (1 - \eta)e_t + \eta x_t y_t. \end{aligned} \tag{2}$$

The plastic neural layers can be either in a feedforward layer (Najarro & Risi, 2020) or part of the recurrent layer (Miconi et al., 2018; 2019).

A challenge for plastic ANNs is the requirement for extensive meta-parameters. For instance, connections with $n_x$ input neurons and $n_y$ output neurons require more than $4n_x n_y$ metaparameters $(A, B, C, D)$, which is even more than the neural connections updated. Rules with fewer meta-parameters such as retroactive neuromodulated plasticity have only been validated in cases where the neural connections are properly initialized (Miconi et al., 2018; 2019).

## 2.4 Implementing Genomics Bottleneck

Large-scale deep neural networks typically lack robustness and generalizability (Goodfellow et al., 2014). A potential way to address this challenge is to adapt a large-scale neural network with relatively simple rules, following the genomics bottleneck in biology. Previous works utilizing genomics bottleneck include encoding

forward, backward rules, and neural connections with a number of tied smaller-scale genomics networks (Koulakov et al., 2021), reducing plasticity rules (Pedersen & Risi, 2021), encoding extensive neural network parameters with pattern producing networks (or hyper-networks) (Stanley et al., 2009; Clune et al., 2009; Ha et al., 2016), and even representing plasticity rules with hyper-networks (Risi & Stanley, 2010; 2012). Among those works, Evolving&Merging (Pedersen & Risi, 2021) is more related to our decomposed plasticity. Based on similar motivations of reducing the learning rules, Evolving&Merging tie plasticity rules of different neural connections based on the similarity and re-evolve the tied rules. However, compared to the decomposed plasticity, it is less biologically plausible, more computationally expensive, and harder to scale up.

## 3  Algorithms

**Problem Settings**. We suppose an agent has its behaviors dependent on both static components (including learning functionalities, and static neural connections) and adaptive components (including plastic neural connections and neuronal states, denoted by $\theta$), which is initially decided by a group of meta-parameters (genotype, denoted by $\phi$, see Figure 2). Notice that in our cases, the adaptive components $\theta$ are always started from scratch, and the meta-parameters $\phi$ only decide the static components; In other cases, the initialization of the adaptive components might also depend on $\phi$ (Miconi et al., 2018; 2019). In *meta training* or outer-loop learning, the meta-parameters $\phi$ are optimized in a set of training tasks $T_j \in \mathcal{T}_{tra}$, and then used for initialization. In *meta testing*, $\phi$ is evaluated on a set of validation / testing tasks $\mathcal{T}_{valid}$ / $\mathcal{T}_{tst}$. For each step in meta-training and meta-testing, the individual *life cycle* of the agent (i.e., the inner loop) refers to the process through which it interacts with environments through observations $i_t$ and actions $a_t$, where $\theta$ is continually updated and change its behaviors. Specifically, in this paper, we mainly consider meta-reinforcement learning problems, where the observation $i_t$ combines current observed states ($o_t$), previous-step action ($a_{t-1}$), and previous-step feedback ($r_{t-1}$) (Duan et al., 2016; Mishra et al., 2018; Wang et al., 2018) (In supervised learning $i_t$ combines the features $x_t$ and the previous-step label $y_{t-1}$ (Santoro et al., 2016)). The inner loop typically has two phases (Beaulieu et al., 2020): The agent first tentatively explores the environment in *meta-training-training* and learns from the observations; It is latterly evaluated in the *meta-training-testing* phase. In *meta-testing*, similarly, the learned meta-parameters are given *meta-testing-training* and *meta-testing-testing* in order. A *life cycle* marks the summarized length of an agent's inner-loop training and testing phases of an agent.

**Decomposed Plasticity**. Considering a plastic layer with pre-synaptic (input) neurons states $\mathrm{x} \in \mathbb{R}^{n_x}$ and post-synaptic (output) neurons states $\mathrm{y} \in \mathbb{R}^{n_y}$, we can rewrite Equation 1 in the matrix form of

$$\Delta W(m, x, y) = m \cdot [W_A \odot (\mathrm{y} \otimes \mathrm{x}) + W_B \odot (\mathbf{1} \otimes \mathrm{x}) + W_C \odot (\mathrm{y} \otimes \mathbf{1}) + W_D], \tag{3}$$

where we use $\odot$ and $\otimes$ to represent "element-wise multiplication" and "outer product", respectively. Here $\Delta W$ is the updates for the neural connections, and $W_A, W_B, W_C, W_D \in \mathbb{R}^{n_y \times n_x}$ are the meta-parameters deciding the learning rules. In decomposed plasticity, we introduce a neuron-dependent decomposition of those meta-parameters, e.g., $W_A = \mathrm{v}_{Ay} \otimes \mathrm{v}_{Ax}$, thus Equation 3 can be changed to

$$\begin{aligned} \Delta W(m, \mathrm{x}, \mathrm{y}) = m \cdot [&(\mathrm{v}_{Ay} \odot \mathrm{y}) \otimes (\mathrm{v}_{Ax} \odot \mathrm{x}) + \mathrm{v}_{By} \otimes (\mathrm{v}_{Bx} \odot \mathrm{x}) \\ &+ (\mathrm{v}_{Cy} \odot \mathrm{y}) \otimes \mathrm{v}_{Cx} + \mathrm{v}_{Dy} \otimes \mathrm{v}_{Dx}], \end{aligned} \tag{4}$$

where $\mathrm{v}_{*,x} \in \mathbb{R}^{n_x}, \mathrm{v}_{*,y} \in \mathbb{R}^{n_y}$. The decomposed plasticity rule contains all $4(n_x + n_y)$ parameters. For large $n_x$ and $n_y$, it is orders of magnitude smaller than the scale of the neural connections ($n_x \times n_y$).

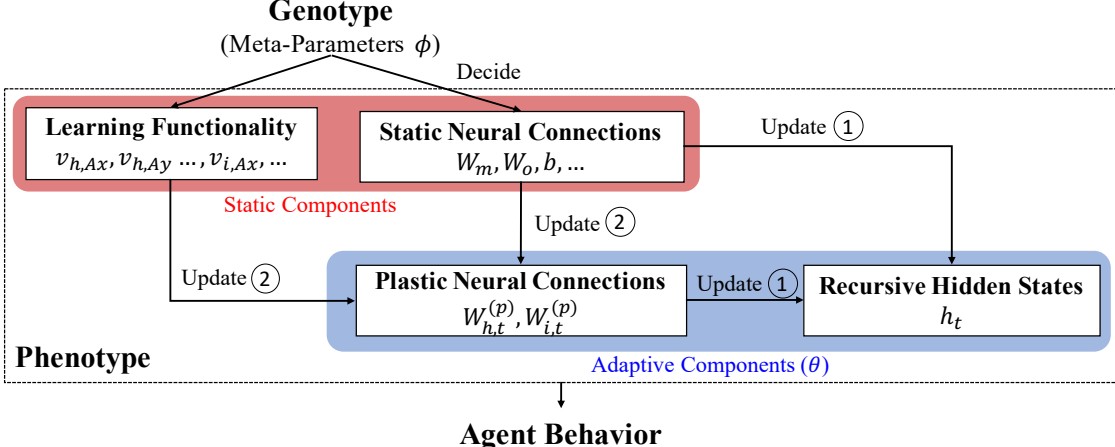

Figure 2: An overview of the modulated plastic RNNs. The static components include the learning rules and static neural connections decided by genotype, which will not change within the inner loop. The adaptive components include the plastic neural connections and hidden neuronal states continually shaped by learning rules and other components.

**Modulated Plastic RNN**. Given a sequence of observations $i_1, ..., i_t, ...$, a plastic RNN updates the hidden states $h_t$ with the following equation:

$$h_{t+1} = tanh(W_{h,t}^{(p)} h_t + W_{i,t}^{(p)} i_t + b), \tag{5}$$

$$a_t = f(W_o h_{t+1}), \tag{6}$$

$$W_{h,t+1}^{(p)} = W_{h,t}^{(p)} + \Delta W(m_{h,t}, h_t, h_{t+1}) \tag{7}$$

$$W_{i,t+1}^{(p)} = W_{i,t}^{(p)} + \Delta W(m_{i,t}, i_t, h_{t+1}). \tag{8}$$

We use superscript $(p)$ to represent plastic neural connections. Unlike previous works of plastic RNN or plastic LSTM that implement plasticity only in $W_{h,t}^{(p)}$, we apply decomposed plasticity for both $W_{h,t}^{(p)}$ and $W_{i,t}^{(p)}$. This further reduces our meta-parameters.

We consider two types of modulation: The pre-synaptic dopamine neuron generates the modulation by a non-plastic layer processing the pre-synaptic neuronal states and sensory inputs; The post-synaptic dopamine neuron integrates the signal from the post-synaptic neuronal states, as follows:

$$\text{Pre-synaptic Dopamine Neuron (\textbf{PreDN}): } m_{h,t}, m_{i,t} = \sigma(W_m[i_t, h_t]) \tag{9}$$

$$\text{Post-synaptic Dopamine Neuron (\textbf{PostDN}): } m_{h,t}, m_{i,t} = \sigma(W_m h_{t+1}) \tag{10}$$

The proposed plasticity can be implemented in both recurrent NNs and forward-only NNs. An overview of plastic RNN is shown in Figure 2. Parameters that decide learning functionality (plasticity rules) and static neural connections are regarded meta-parameters ($\phi$). The variables contained by plastic neural connections and recursive hidden states are regarded adaptive components in the phenotype ($\theta$). The adaption of the model is achieved through updating the hidden states (Update ①, or Equation 5) and updating the plastic neural connections (Update ②, or Equation 7,8). Figure 2 can fit various model-based and plasticity-based learning algorithms. Based on this setting, the genomic bottleneck indicates $|\theta| \gg |\phi|$.

**Outer-Loop Evolution**. Given task $T_j \in \mathcal{T}$, by continually executing the inner loop including *meta-training-training* and *meta-training-testing*, we acquire the fitness of the genotype (meta-parameters) $\phi$ at the end of its life cycle, denoted as $Fit(\phi, T_j)$. The genotype can be updated using an evolution strategies (ES) approach,

e.g. Rechenberg (1973); Salimans et al. (2017):

$$\phi^{k+1} = \phi^k + \alpha \frac{1}{g} \sum_{i=1}^{g} Fit(\phi^{k,i}, T_k)(\phi^{k,i} - \phi^k). \tag{11}$$

The superscript $k$ and $i$ represent the $k$th generation and the $i$th individual in that generation. The subscript $\tau$ marks the duration of an individual life cycle. The population $\phi^{k+1,i}$ is sampled around $\phi^{k+1}$ with the covariance matrix $C = \sigma^2 \mathbb{I}$. For high-dimensional meta-parameters, selecting proper hyperparameters (e.g. the covariance matrix $C$) is nontrivial. Improper selection could end up in inefficient optimization and local optimums. To address this challenge, CMA-ES greatly improves optimization efficiency and robustness by automatically adapting the covariance matrix using the evolution path (Hansen & Ostermeier, 2001). However, it comes at the price of increasing the per-generation computational complexity from $\Theta(|\phi|)$ to $\Theta(|\phi|^2)$, which is infeasible for large-scale ANNs. Alternatively, we use seq-CMA-ES (Ros & Hansen, 2008) where the covariance matrix degenerates to $\Theta(|\phi|)$ by preserving the diagonal elements of $C$ only, which is affordable and empirically more efficient compared to ES.

The proposed model can also be optimized with a gradient-based optimizer following Miconi et al. (2018). In cases of supervised training, ES is typically less efficient than gradient-based optimization. However, for meta-RL with sparse rewards, ES could be more efficient than gradient-based optimizers (Salimans et al., 2017). Moreover, considering the models obeying the genomics bottleneck ($|\theta| \gg |\phi|$) and long life cycles ($\tau \gg 1$), ES-type optimizers could be a potentially more economical choice in both CPU/GPU memory and computation consumption.

**Biological Plausibility**. The decomposed plasticity is inspired by neuronal differentiation (Morrison, 2001) in biological systems. Instead of assuming each neural connection has unique learning rules, it might be more biologically plausible to propose that the learning rules are related to pre-synaptic and post-synaptic neurons separately. Although there are other ways to make the learning rules more compact, e.g., hyper-networks (Risi & Stanley, 2012), and Evolving&Merging (Pedersen & Risi, 2021), the decomposed plasticity is relatively straightforward and easier to implement by tensor operations. The effect of neuromodulations on long-term memory and learning in biological systems is well supported by experiments (Schultz, 1997; Shohamy & Adcock, 2010). Typical neural modulated plastic ANNs assign a unique modulator for a neural connection or a neuron (Soltoggio et al., 2008; Miconi et al., 2019), which is inconsistent with the fact that dopamine neurons are much fewer than the other neurons in biological systems (German et al., 1983). Our experiments show that assigning only a single dopamine neuron for an entire plastic layer can yield satisfying performance, saving many meta-parameters.

**Summary**. We formalize the inner-loop learning and meta-training process in Algorithm 1 and Algorithm 2. The framework [1] are also applicable to the other model-based and plasticity-based meta-learning. Note that in meta-RL, a life cycle includes multiple episodes. The agent must gain experience from the earlier episodes (meta-training-training) to perform well in later episodes (meta-training-testing). Therefore, we use the variable $w_z$ to tune the importance of each episode to assess fitness. An explanation of these training settings can be found in the Appendix A.2.

## 4 Experiments

### 4.1 Experiment Settings

We validate the proposed method in MetaMaze2D (Wang, 2021), an open-source maze simulator that can generate maze architectures, start positions, and goals at random. The observation $i_t$ is composed of three parts: the $3 \times 3$ grids observed ($o_t$), the previous-step action ($a_{t-1}$), and the previous-step reward ($r_{t-1}$). The maze structures, their positions, and the goals are hidden from the agents. Our settings have a 15-dimension input and a 5-dimension output in all. The output action includes 4 dimensions deciding the probability of taking a step in its four directions (east, west, south, north) and one additional dimension deciding whether it will take a softmax sampling policy or an argmax policy. On top of the plastic layers, we add a non-plastic

---

[1]source code available at `https://github.com/WorldEditors/EvolvingPlasticANN`

---

**Algorithm 1** Inner-Loop Learning

---

1: Input $\phi$, $\mathcal{T}$.
2: **for** $T \in \mathcal{T}$ **do**
3:     Reset $\theta$ to scratch.
4:     **for** $z = 0, 1, 2, ...$ until the end of a life cycle **do**
5:         **for** $t = 0, 1, 2, ...$ until the end of an episode **do**
6:             Observe $o_t$, set $i_t = [o_t, a_{t-1}, r_{t-1}]$.
7:             Update $\theta$ using $\phi$ and Equation 4,5,6,7,8, acquire output $a_t$.
8:             Execute $a_t$, receive $r_t$.
9:         $R_z = \sum_t r_t$
10:     $Fit(\phi, T) = \sum_z w_z \cdot R_z$.
11: Return $Fit(\theta_{\text{Gene}}, \mathcal{T}) = \frac{1}{|\mathcal{T}|} \sum_{T \in \mathcal{T}} Fit(\phi, T)$.

---

**Algorithm 2** Meta-Training and Meta-Testing

---

1: Pre-sample $\mathcal{T}_{\text{valid}}$ and $\mathcal{T}_{\text{tst}}$.
2: Randomly sample $g$ initial genotypes $\phi^{0,i}$, $i = 1, ..., g$.
3: **for** Generations $k = 0, 1, 2, ...$ until convergence **do**
4:     Randomly sample training tasks $\mathcal{T}_{\text{tra}}$.
5:     **for** $i = 1, 2, ..., g$ **do**
6:         Acquire average fitness $Fit(\phi^{k,i}, \mathcal{T}_{\text{tra}})$ by calling Algorithm 1
7:     Apply Seq-CMA-ES to acquire the next generation centroid $\phi^{k+1}$ and population $\phi^{k+1,i}$
8:     Acquire $Fit(\phi^k, \mathcal{T}_{\text{valid}})$ by Algorithm 1, record $\phi^*$ acquiring the best fitness.
9: Return $\phi^*$, $Fit(\phi^*, \mathcal{T}_{\text{tst}})$.

---

output layer that processes the hidden units to 5-dimensional output. We found that the performance was lower if the output layer was plastic, and since it contains relatively few parameters, keeping it static does not violate the genomics bottleneck. The agents acquire the reward of 1.0 by reaching the goal and $-0.01$ in other cases. Each episode terminates when reaching the goal, or at the maximum of 200 steps. A life cycle has totally 8 episodes.

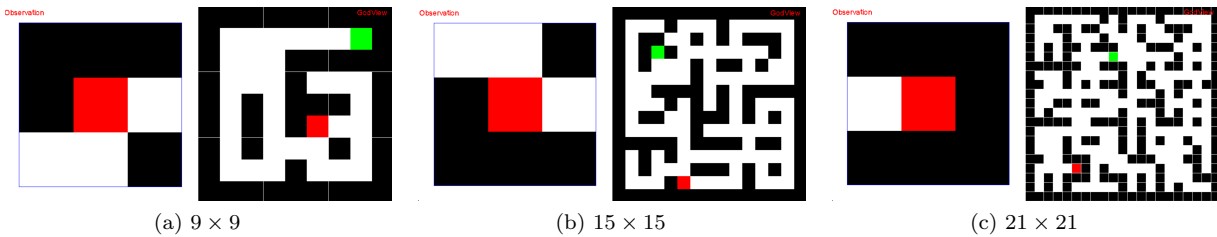

|          (a) $9 \times 9$          |          (b) $15 \times 15$          |          (c) $21 \times 21$          |

Figure 3: Cases of mazes of different scales including the observed states ($o_t$) and the god view. The red squares mark the current positions of the agents; The green squares mark the goals.

For meta-training, each generation includes $g = 360$ genotypes evaluated on $|\mathcal{T}_{\text{tra}}| = 12$ tasks. The genotypes are distributed to 360 CPUs to execute the inner loops. The variance of the noises in Seq-CMA-ES is initially set to be 0.01. Every 100 generations we add a validating phase by evaluating the current genotype in $|\mathcal{T}_{\text{valid}}| = 1024$ (*validating tasks*). By reducing $g$ or $|\mathcal{T}_{\text{tra}}|$ we observed obvious drop in performances. Scaling up those settings will stabilize the training but lead to increase of time and computation costs. Meta training goes for at least 15,000 generations, among which we pick those with the highest validating scores for meta-testing.

The testing tasks include $9 \times 9$ mazes (Figure 3 (a)), $15 \times 15$ mazes (Figure 3 (b)), and $21 \times 21$ mazes (Figure 3 (c)) sampled in advance. There are $|\mathcal{T}_{\text{tst}}| = 2048$ tasks for each level of mazes. We run meta-training of the

compared method twice, from each of which we pick those with the highest fitness in validating tasks. We report the meta-testing results on the two groups of selected meta-parameters. The convergence curves of meta-training is left to Appendix A.3.2.

We include the following methods for comparison:

- **DNN**: Evolving the parameter of a forward-only NN with two hidden fully connected layers (both with a hidden size of 64) and one output layer. Two different settings are applied: In DNN, we only use the current state as input; In Meta-DNN, we concatenate the state and the previous-step action and feedback as input.

- **Meta-RNN**: Employing RNN to encode the observation sequence, the parameters of RNN are treated as meta-parameters. We evaluate the hidden sizes of 8 (Meta-RNN-XS), 16 (Meta-RNN-S), and 64 (Meta-RNN).

- **Meta-LSTM**: Employing LSTM to encode the observation sequence, the parameters of LSTM are treated as meta-parameters. We evaluate the hidden sizes of 8 (Meta-LSTM-XS), 16 (Meta-LSTM-S), and 64 (Meta-LSTM).

- **PRNN**: Applying the $\alpha ABCD$ plasticity rule (Equation 3) to the PRNN. We also evaluate the hidden sizes of 8 (PRNN-XS), 16, (PRNN-S), and 64 (PRNN).

- **DecPDNN**: Applying the decomposed plasticity to the first two layers of Meta-DNN.

- **DecPRNN**: Applying the decomposed plasticity to PRNN (Equation 4). We evaluate the hidden sizes of 32 (DecPRNN-S) and 64 (DecPRNN).

- **Retroactive** PRNN : Applying the retroactive neuromodulated plasticity (Equation 2) to PRNN, but only to the recursive connections ($W_{h,t}^{(p)}$), the input connections ($W_{i,t}^{(p)}$) are not included. Following Backpropamine (Miconi et al., 2019), the initial parameters of the connection weights are not from scratch, but decided by meta-parameters.

- **Retroactive(Random)** PRNN: Start the plastic neural connections from scratch in Miconi et al. (2019).

- **Evolving&Merging**: Implementing evolving and merging (Yaman et al., 2021) in PRNN, where we start training with the $\alpha ABCD$ rules and reduce those rules using K-Means clustering and re-train the tied rules. But unlike the original proposal that evolves and merges multiple times, we merge and re-evolve for only one time, reducing the 20224 rules to 1144 rules, the same as the scale of meta-parameters in DecPRNN.

Notice that the plastic neural networks may be further combined with different types of neural modulation, including non-modulation, PreDN (Equation 9), and PostDN (Equation 10). We compare different modulations in DecPRNN. For all the other plasticity-based methods, we apply PostDN, which is proven to be state-of-the-art. To emphasize the impact of different meta-parameters and adaptive components, we show the number of meta-parameters in genotypes and the number of the variables in adaptive components of all the compared methods in Figure 4.

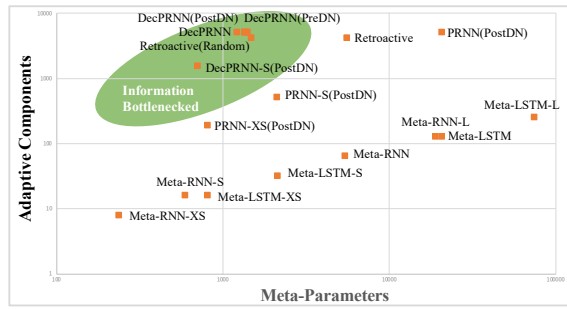

Figure 4: Over-viewing the number of variables in static and adaptive components of various methods.

### 4.2 Experiment Results

#### 4.2.1 Best-Rollout Performances

We first show the best-rollout performance of the compared methods, which is calculated by picking the rollout with the highest average rewards among their 8-rollout life cycles (Figure 5). Besides the rewards, we

also show the *failure rate*, which marks the ratio of mazes among $\mathcal{T}_{tst}$ where the agent fails to reach the goal within 200 steps. The best-rollout performances are regarded as an indicator of the agents' learning potential. We evaluated all the compared methods in $9 \times 9$ and $15 \times 15$ mazes and only the most representative and competitive methods in $21 \times 21$. Here we omit all the results of DNN and Meta-DNN (see Appendix A.3.1) as showing those bars might cover up the other differences.

**Plasticity-based vs. Model-based Learners**. Generally, we show that plasticity-based learners perform better than model-based learners (Meta-RNN, Meta-LSTM), especially for more complex cases of larger mazes. For instance, in $9 \times 9$ mazes, the best model-based learner (Meta-LSTM) is slightly better than the best plasticity-based learner (DecPRNN(PostDN)). However, they underperform plasticity-based learners in $15 \times 15$ mazes and $21 \times 21$ mazes. Among model-based learners, Meta-LSTMs are better than Meta-RNNs, which is also more obvious in complex cases. For both Meta-LSTMs and Meta-RNNs, a hidden size below 16 yields a clear decline in the performance, while a larger scale generally results in better performances. Since larger mazes mean a longer life cycle (see Appendix A.3.3), our results show that plasticity-based learners are more powerful in long-term memorization. Notice that model-based learners generally have smaller-scale adaptive components (see Figure 4 or Table 1) but larger-scale meta-parameters compared with plasticity-based learners. We argue that the scale of adaptive components also plays an important role in memorization.

**Plasticity Rules Comparison**. We validate that retroactive PRNN with PostDN can improve the Meta-RNNs, which is consistent with Miconi et al. (2019). However, when the connection weights are randomly initialized, retroactive PRNN performs much poorer than the other plasticity rules. It is not surprising as the retroactive plasticity have the fewest learning rules. In contrast, the $\alpha ABCD$ rule (PRNN), Evolving&Merging, and decomposed plasticity rule can do reasonably well. For standard hidden size (64), decomposed plasticity can do the best among all plasticity rules, even though theoretically, the upper bound of $\alpha ABCD$ rule should be higher. It might be related to the meta-training process since a larger meta-parameter space raises higher challenge for ES. Another possible cause can be larger meta-parameters requires larger $|\mathcal{T}_{tra}|$ to avoid overfitting specific task distributions. The Evolving&Merging can not do as well as decomposed plasticity in $9 \times 9$ and $15 \times 15$ mazes. Considering it has the same scale of meta-parameters as decomposed plasticity, and it requires at least two stages of training (meta-train PRNN in the first stage, then switch to merged rules in the second stage), the Evolving&Merging seems less attractive than decomposed plasticity. Another surprising fact is that PRNN-S with only 16 hidden units can do very well in $9 \times 9$ and $15 \times 15$ mazes and reasonably well in $21 \times 21$ mazes. It has slightly larger-scale meta-parameters compared with DecPRNN (2119 vs. 1347). The scale of its adaptive components is much smaller than PRNN or DecPRNN (512 vs. 5120) but still much larger than model-based learners. It seems that the reduction of meta-parameters plays a crucial role here. However, as we tested DecPRNN-S ($|\theta| = 1536$, $|\phi| = 707$), and even smaller PRNN-XS ($|\theta| = 192$, $|\phi| = 809$), the performances is significantly degraded. Based on these findings and considering retroactive PRNN's performance and model-based learners' performance, we may conclude that relatively smaller-scale meta-parameters and larger-scale adaptive variables are helpful but with boundaries.

**Effect of Neural Modulations**. The results also show that PostDN > PreDN > non-modulation for DecPRNN. The difference between modulated/non-modulated DecPRNN is nontrivial, showing that employing even very few neural modulators can be crucial. The success of PostDN over PreDN suggests that the modulation signal is better to "backpropagate" than to integrate sensory inputs. Inspired by the analogy between rewards and neural modulations (Schultz, 1997), this seems to support the functionality of intrinsic reward and intrinsic motivation.

### 4.2.2 Inner-Loop Visualization

We plot the per-rollout rewards and failure rates within the agent's life cycles of several competitive methods in Figure 6. The performances of model-based and plasticity-based learners deviate over time: The model-based learners typically start at a good level but stop improving at $3^{rd}$ rollouts and suffer from a relatively low ceiling. In contrast, the plasticity-based learner starts from scratch but keeps improving through its life cycle to eventually a higher level. Also, we observe the different levels of improvement in different mazes. For simpler $9 \times 9$ mazes, most methods stop improving after their $3^{rd}$ rollouts. For complex $21 \times 21$ mazes, we can see signs of improvement even at the last rollouts regarding the best plasticity-based learners.

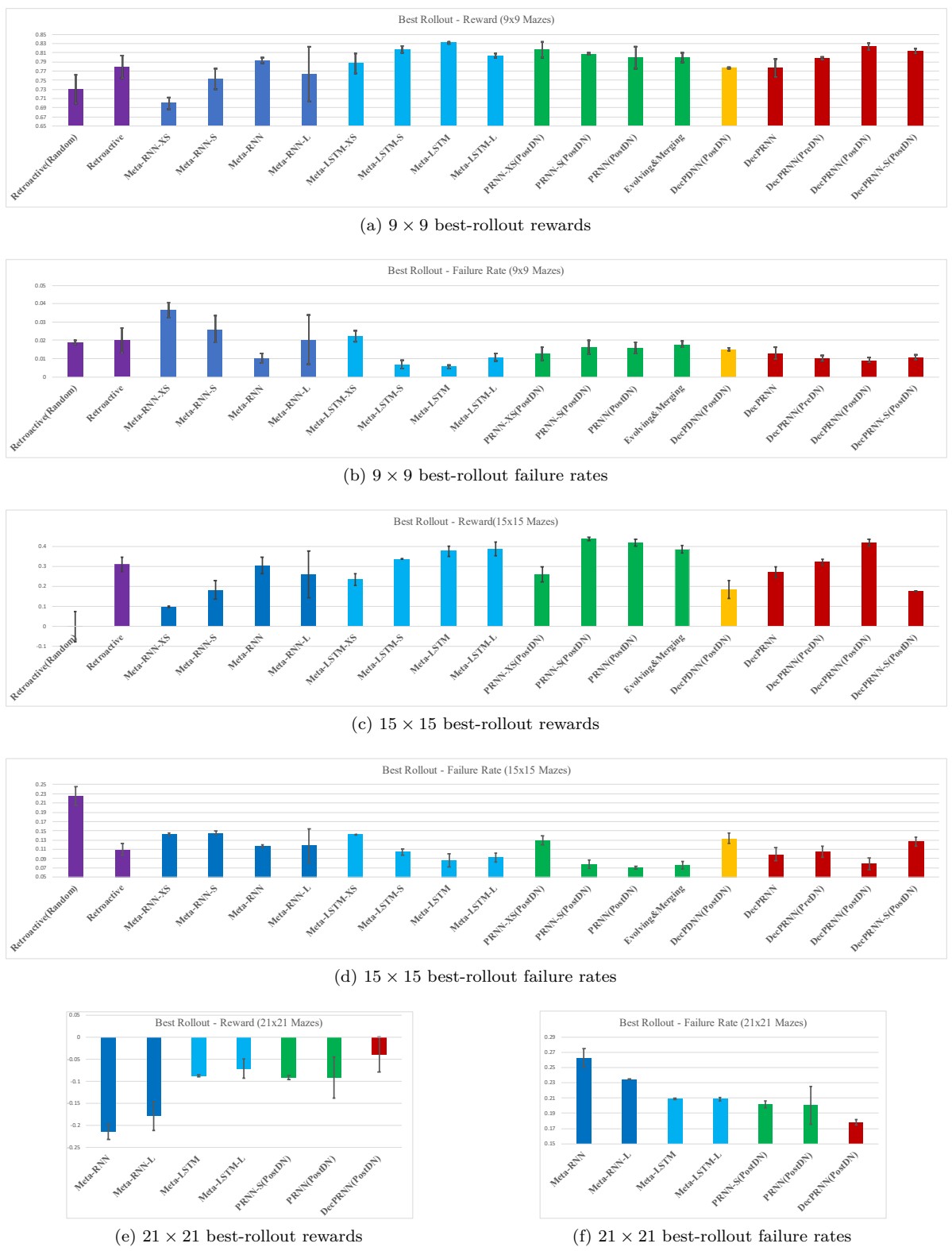

(a) $9 \times 9$ best-rollout rewards

(b) $9 \times 9$ best-rollout failure rates

(c) $15 \times 15$ best-rollout rewards

(d) $15 \times 15$ best-rollout failure rates

(e) $21 \times 21$ best-rollout rewards

(f) $21 \times 21$ best-rollout failure rates

Figure 5: Best-rollout performances (including the rewards and failure rates) of compared methods. Methods of the same genre with different scales or different modulations are marked with the same color, while different genres of methods are marked with different colors.

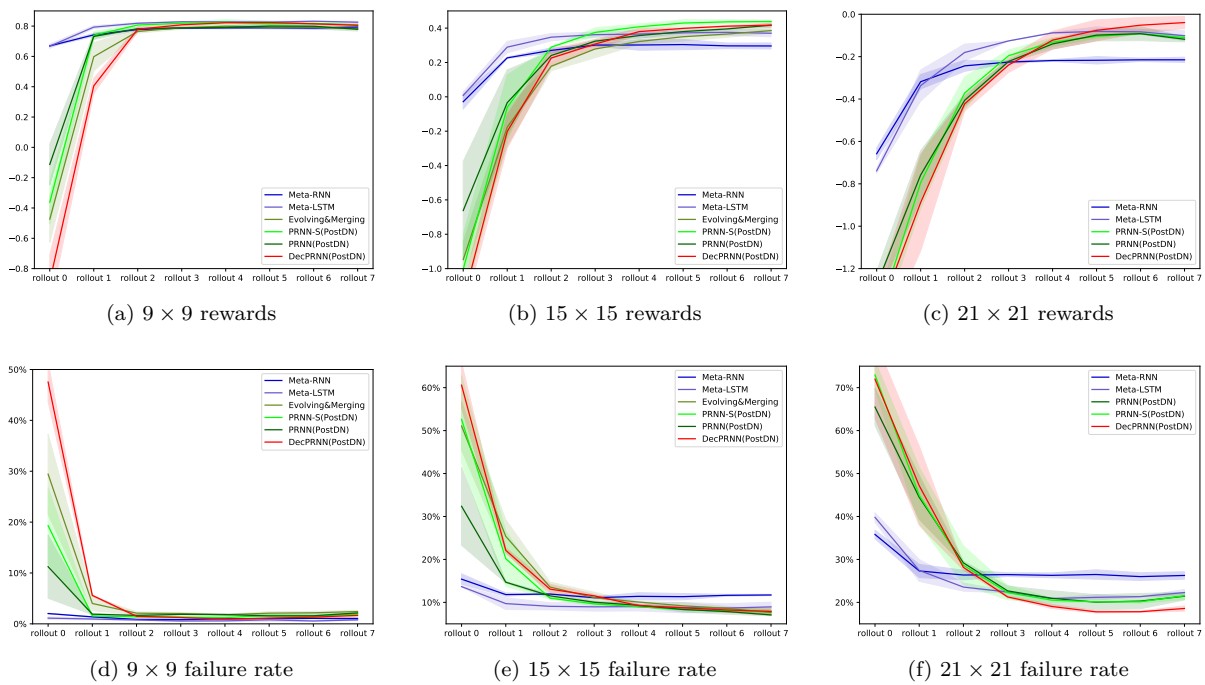

Figure 6: Per-rollout performances of selected methods for 8-rollouts life cycles.

**Visualizing Adaptive Components**. We present the development of the adaptive components within the agents' life cycles, including hidden states $(h_t, c_t)$ and the plastic neural connection weights $(W_{h,t}^{(p)})$. We first sample 3 different tasks. For each task, we run the life cycle of different methods twice, which yields 6 trajectories of adaptive variables for each method. Since those tensors are in relatively high dimensions, we run t-SNE visualization (Van der Maaten & Hinton, 2008) to map them to a 2-D space and show their temporal traces in Figure 7. In Figure 7(a), we also show the correspondence of the traces to the actual trajectories in the maze. Comparing Figure 7(a) and (b), we see that the plastic neural connection weights behave differently from hidden states: The connection weights smoothly migrate toward certain directions across the agent's life, the traces of which are correlated with the hidden task configuration (Figure 7(a,c,d,e,f)); In contrast, the hidden states vibrate fast without a clear sign of long-term orientation (Figure 7(b, g,h,f)). It could possibly be used to explain why plasticity-based learners do better than model-based learners in long-horizon tasks and why combining recursion and plasticity can yield better performances: PRNN and DecPRNN can keep the long-range information in connection weights and leave the short-term information to hidden states, while recursion-only learners must keep all those memories in the hidden states, where they might interfere with each other. In Figure 7(a) we also mark the possible "sweet spots" of the plastic neural connections for the three tasks, where the agents find the optimal solutions and the connection weights are at convergence. In other cases, especially non-modulated DecPRNN (Figure 7(c)) and recursion-free DecPDNN(PostDN) (Figure 7(d)), those "sweet spots" can be hardly observed, suggesting that their learning processes might be noisier.

To quantitatively investigate the learning process of plastic neural connections and hidden states, we introduce two measures related to short-term and long-term behaviors, respectively. To measure the short-term vibration of the learning process of variable $x_t$ (which can be either connection weights $W_t^{(p)}$ or hidden states $h_t$), we introduce $\sigma_{vib}(x_t) = \mathbb{E}[|x_t - \hat{x}_t|]$, where $\hat{x}$ is the moving average of $x$ with a sliding window $[t-m, t+m]$ (here we use $m = 7$, window size of 15), $|\cdot|$ denotes the L2-norm. To measure the long-term migration of $x_t$, we use $d_{mig}(x_t) = \mathbb{E}[|\hat{x}_t - \hat{x}_0|]$. We show $\sigma_{vib}$ and $d_{mig}$ against the steps of the agent's life cycle in Figure 8(a) and (b) from the statistics of testing on the full 2048 $15 \times 15$ mazes. Since 90% of the life cycles are within 1000 steps (see Appendix A.3.3), we show only the statistics within 1000 steps. We can see more

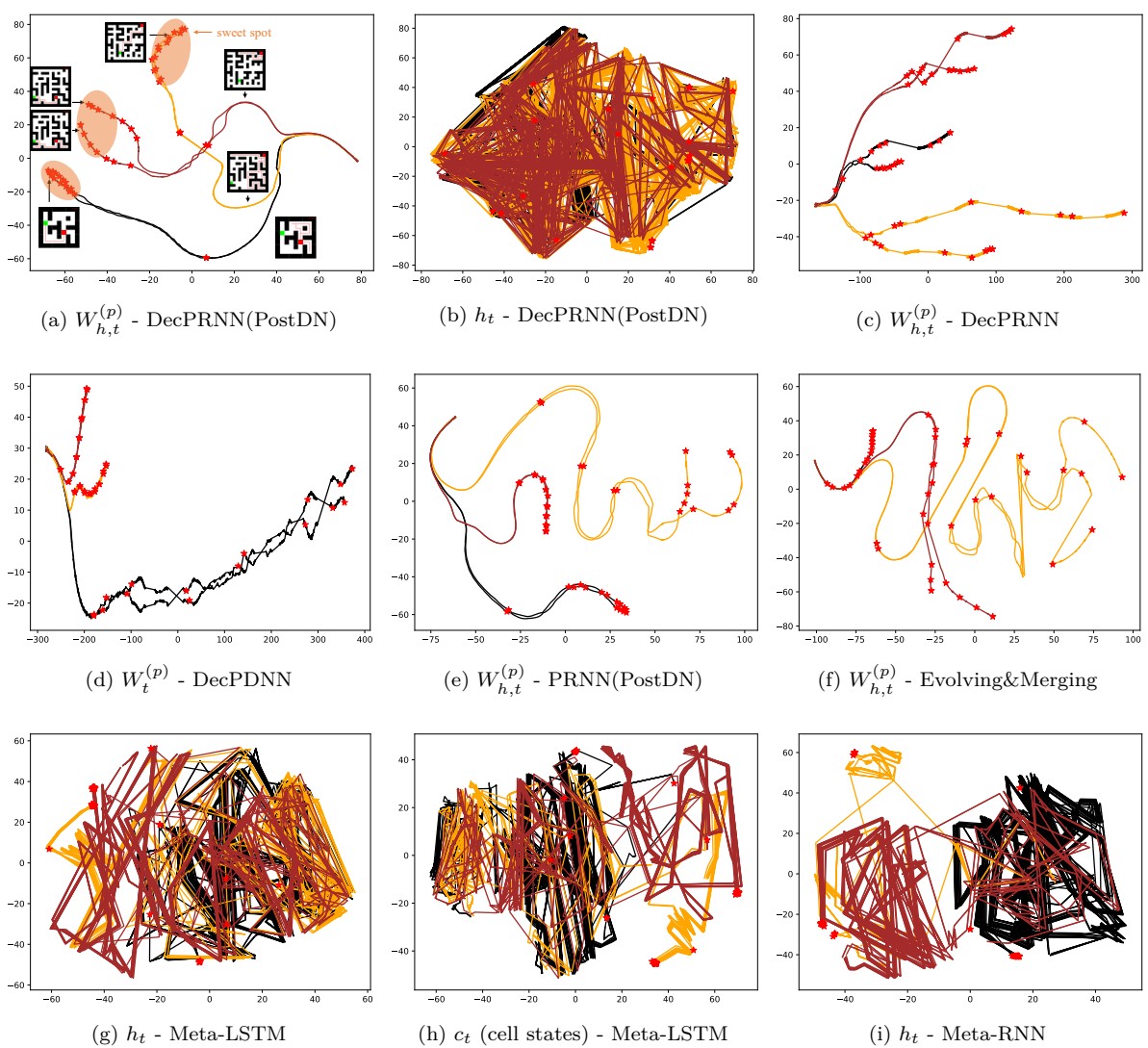

Figure 7: t-SNE visualization of the transformation of the connection weights ($W_{h,t}^{(p)}$) and hidden states ($h_t$) in various methods. Each figure represents the traces of particular components in a specific method. Each method is demonstrated in 3 different tasks twice, yielding 6 traces, as shown in (a), where each color (brown, yellow, black) corresponds to a unique task. The red ★ in each trace marks the end of a rollout. There are 8 ★s in each trace corresponding to 8 rollouts in each life cycle.

clearly that the plastic neural connections migrate much more in the long run ($d_{mig}$) compared with the hidden states, which are nearly flat within the life cycle. In contrast, the hidden states have more significant short-term vibrations ($\sigma_{vib}$) than the plastic connection weights. Moreover, DecPDNN has a larger-scale $\sigma_{vib}(W_t^{(p)})$ and $d_{mig}(W_t^{(p)})$ in the later of the life cycle, which might be attributed to a lack of hidden states to contain short-term memories. The non-modulated DecPRNN also has a larger scale in both $\sigma_{vib}(W_{h,t}^{(p)})$ and $d_{mig}(W_{h,t}^{(p)})$, validating that the neural modulation plays crucial role in regulating the learning of neural connections.

**Analyses on Explorations**. It is well known that reinforcement learning depends on exploration and exploitation simultaneously. We are then interested in investigating whether the inner loop has learned to balance exploration and exploitation. We visualize the exploration of the agents by using the *coverage*

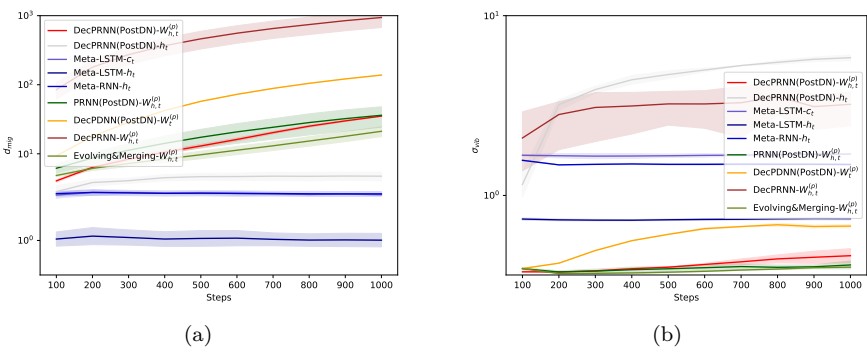

(a)                    (b)

Figure 8: Quantitative evaluation of the long-term and short-term change of the adaptive variables evaluated in $15 \times 15$ mazes. (a) $d_{mig}(W_t^{(p)})$ or $d_{mig}(h_t)$ measuring the long-term migration. (b) $\sigma_{vib}(W_t^{(p)})$ or $\sigma_{vib}(h_t)$ measuring the short-term vibration.

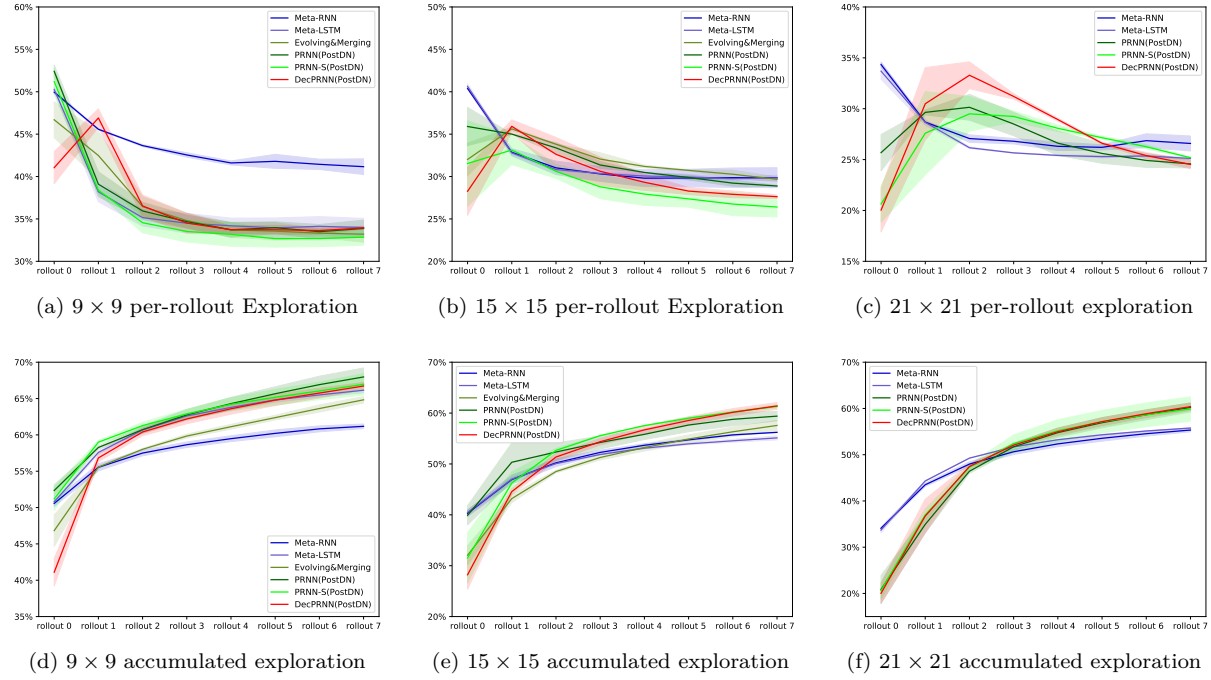

(a) $9 \times 9$ per-rollout Exploration   (b) $15 \times 15$ per-rollout Exploration   (c) $21 \times 21$ per-rollout exploration

(d) $9 \times 9$ accumulated exploration   (e) $15 \times 15$ accumulated exploration   (f) $21 \times 21$ accumulated exploration

Figure 9: The coverage rate of the visited locations showing the exploration of different agents in various mazes.

*rate* of the mazes, which is the unique locations the agent visited divided by all the reachable locations in that maze. We plot both the per-rollout coverage rate, and the accumulated coverage rate (by counting the uniquely visited locations since the beginning of its life cycle) of meta-testing in Figure 9. There are several interesting points worth mentioning. First, we see that all model-based and plasticity-based learners learn to explore more at the beginning (meta-testing-training stage) and gradually reduce the exploration. Second, some plasticity-based methods, especially those obeying the genomics bottleneck (Evolving&Merging and DecPRNN(PostDN)), are more deterministic in their first rollouts but then explore more in the 2nd and $3^{rd}$ rollout. It is also consistent with Figure 7(a), where we show the traces of neural connections in DecPRNN(PostDN) for different tasks highly overlapped in the very beginning. A tentative explanation to this strange behavior is that the agent is experiencing some " warm-up stage" before actually learning about

the maze since its initial neural connections can not support any complex behavior. Third, we find that by summing up the per-rollout explorations, the plasticity-based learners do not have the largest exploration every rollout but have the highest accumulated exploration. It could possibly mean that plasticity-based learners are more efficient in designing cross-episode exploration strategies.

## 5 Limitations and Future Prospects

Currently, typical large-scale deep models work with mostly static components and few adaptive components. They have been powerful in pre-defined tasks but suffered from high customization costs and the inability to generalize to various scenarios with little human effort. In this paper, we suggest designing models with relatively larger-scale adaptive components with inspiration from plasticity and the genomics bottleneck. Those models are expected to be not necessarily capable of everything initially, but capable of learning to accomplish very complex tasks by interacting with the environments and learning automatically.

There are several limitations to this work. First, although our environments have probably been the most challenging 2D maze tasks that have been addressed until now, it is still far too simple compared with many tasks in reality. It also needs to be validated in image-related studies. We believe that building diverse and close-to-reality simulators is essential for the future development of AI. Although a lot of effort is devoted to simulators where agents can learn general locomotion skills (Dosovitskiy et al., 2017; Yu et al., 2020), natural language understanding (Hermann et al., 2017; Yu et al., 2018; Chevalier-Boisvert et al., 2018), natural language generation (Havrylov & Titov, 2017), and even the construction of artifacts (Grbic et al., 2021), there is still a long way to go, given the huge gap between simulation and reality. Second, we believe that our outer-loop optimizer (the Seq-CMA-ES) is a good choice at the current stage, but it still severely limits the model design. Plasticity rules will be more prospective if combined with more sophisticated models, but the meta-training turns out to be the bottleneck. Meanwhile, the genomics bottleneck can alleviate this problem since it means fewer meta-parameters and thus less burden on meta-training, but it is not without limits. To build valuable AIs for reality might still cost millions of meta-parameters calculated on inner loops of millions of steps and maintaining hundreds of millions of adaptive variables. Third, we have mainly tested our settings in a meta-learning framework. An essential setting will be enabling a single agent to continually adapt to nonstationary tasks during its lifetime without forgetting those old ones (Beaulieu et al., 2020). This should be a direction for future investigations.

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

## A Appendix

### A.1 Details of The Model Structures

All the methods compared have a static output layer mapping the hidden units to 5-dimension actions. The difference lies in the rest part that maps the inputs to the hidden units. We list the meta-parameters and adaptive components of all the methods compared in Table 1.

Table 1: Model Structures of Compared Methods

|  | Hidden Units | Adaptive Variables ($|\theta|$) | Meta Parameters ($|\phi|$) |
| --- | --- | --- | --- |
| **DNN** | 64+64 | 0 | 5,125 |
| **Meta-DNN** | 64+64 | 0 | 5,509 |
| **Meta-RNN-XS** | 8 | 8 | 237 |
| **Meta-RNN-S** | 16 | 16 | 597 |
| **Meta-RNN** | 64 | 64 | 5,445 |
| **Meta-RNN-L** | 128 | 128 | 19,077 |
| **Meta-LSTM-XS** | 8 | 16 | 813 |
| **Meta-LSTM-S** | 16 | 32 | 2,133 |
| **Meta-LSTM** | 64 | 128 | 20,805 |
| **Meta-LSTM-L** | 128 | 256 | 74,373 |
| **Evolving&Merging** | 64 | 5,120 | 1,347 |
| **Retroactive** | 64 | 4,160 | 5577 |
| **Retroactive(Random)** | 64 | 4,160 | 1481 |
| **PRNN-XS(PostDN)** | 8 | 192 | 809 |
| **PRNN-S(PostDN)** | 16 | 512 | 2,119 |
| **PRNN(PostDN)** | 64 | 5,120 | 20,743 |
| **DecPDNN(PostDN)** | 64+64 | 5,056 | 1,411 |
| **DecPRNN** | 64 | 5,120 | 1,217 |
| **DecPRNN(PreDN)** | 64 | 5,120 | 1,379 |
| **DecPRNN-S(PostDN)** | 32 | 1,536 | 707 |
| **DecPRNN(PostDN)** | 64 | 5,120 | 1,347 |

### A.2 Meta-Training Settings

To calculate the fitness of an agent regarding its life cycle of 8 rollouts, we apply $Fit(\phi, T) = \sum_z w_z \cdot R_z$, with $w_z$ be

$$w_z = \begin{cases} 0 & z < 2 \\ 0.80^{(\tau - z - 1)} & else \end{cases}$$

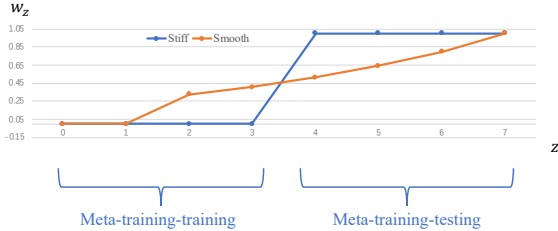

Figure 10: "stiff" and "smooth" training and testing phases in meta-learning.

Here $\tau = 8$ denotes the episodes in the life cycle. This formula is slightly different from canonical meta-supervised-learning settings, where $w_z = 0$ for meta-training-training (or support set) and $w_z = 1$ for

meta-training-testing (or query set). The canonical training and testing phase settings guide the meta-training to improve the performance on the query set only. Although this "stiff" setting can be extended to meta-RL Finn et al. (2017), it is actually more difficult to decide the boundary between meta-training-training and meta-training-testing in meta-RL since we want the agent to gradually improve its performance over its lifetime. Thus, we propose this "smooth" setting for the training and testing phases in the inner loop (Figure 10). In our cases, we found it to be more effective than "stiff" settings. For the outer-loop optimizer (seq-CMA-ES), we used an initial step size of 0.01, and the covariance $C = \mathbb{I}$ for all the compared methods.

### A.3 Additional Results

#### A.3.1 Performances of DNN and Meta-DNN

Table 2: Best-Rollout performance (rewards) of DNN and Meta-DNN, along the performances of random policy and oracles.

|  | Maze $9 \times 9$ | Maze $15 \times 15$ | Maze $21 \times 21$ |
|---|---|---|---|
| **DNN** | $-0.860 \pm 0.038$ | $-1.603 \pm 0.025$ | |
| **Meta-DNN** | $0.480 \pm 0.070$ | $-0.506 \pm 0.018$ | |
| **Random** | $-1.308 \pm 0.031$ | $-1.934 \pm 0.010$ | $-1.976 \pm 0.005$ |
| **Oracle** | $0.908$ | $0.820$ | $0.781$ |

#### A.3.2 Convergences of Meta-Training

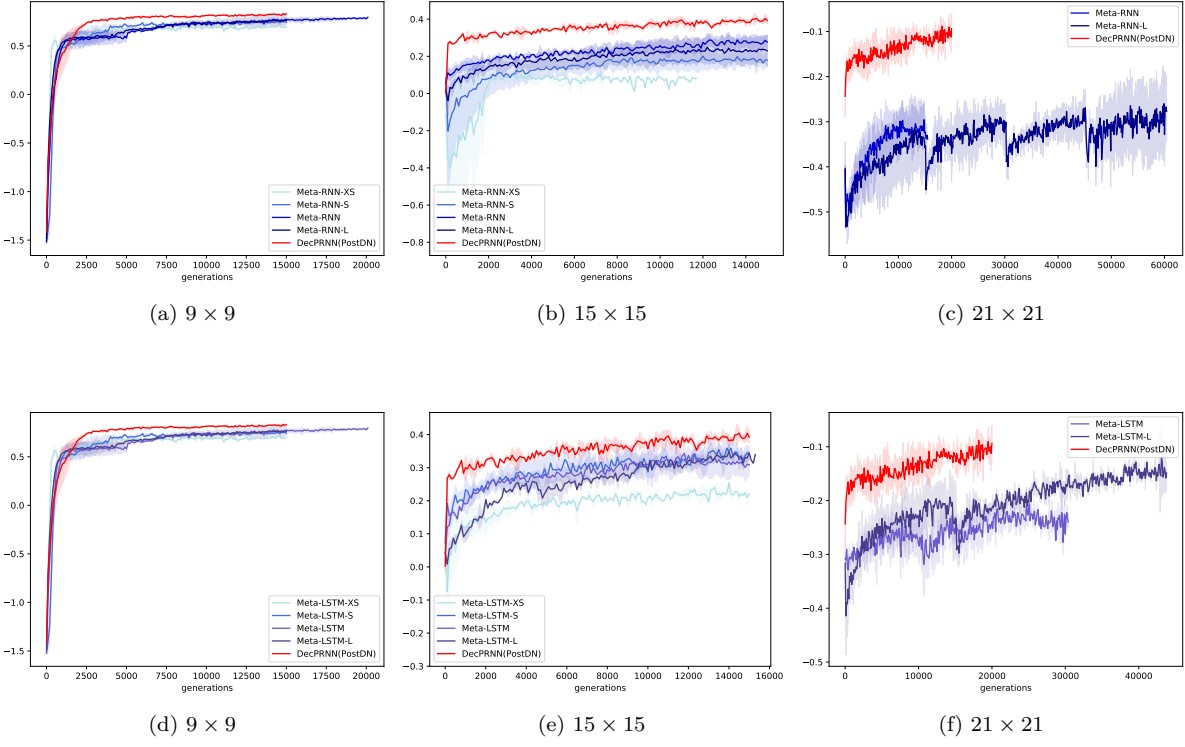

We present the average fitness score evaluated on validating tasks ($\mathcal{T}_{valid}$) against the evolved generations of all the compared methods in Figure 9. For clarity, we group model-based and plasticity-based learners into 6 groups and show each group independently to avoid confusion. For comparison, we also add DecPRNN(PostDN) in each group. Notice that we follow a curriculum learning process for the meta-training.

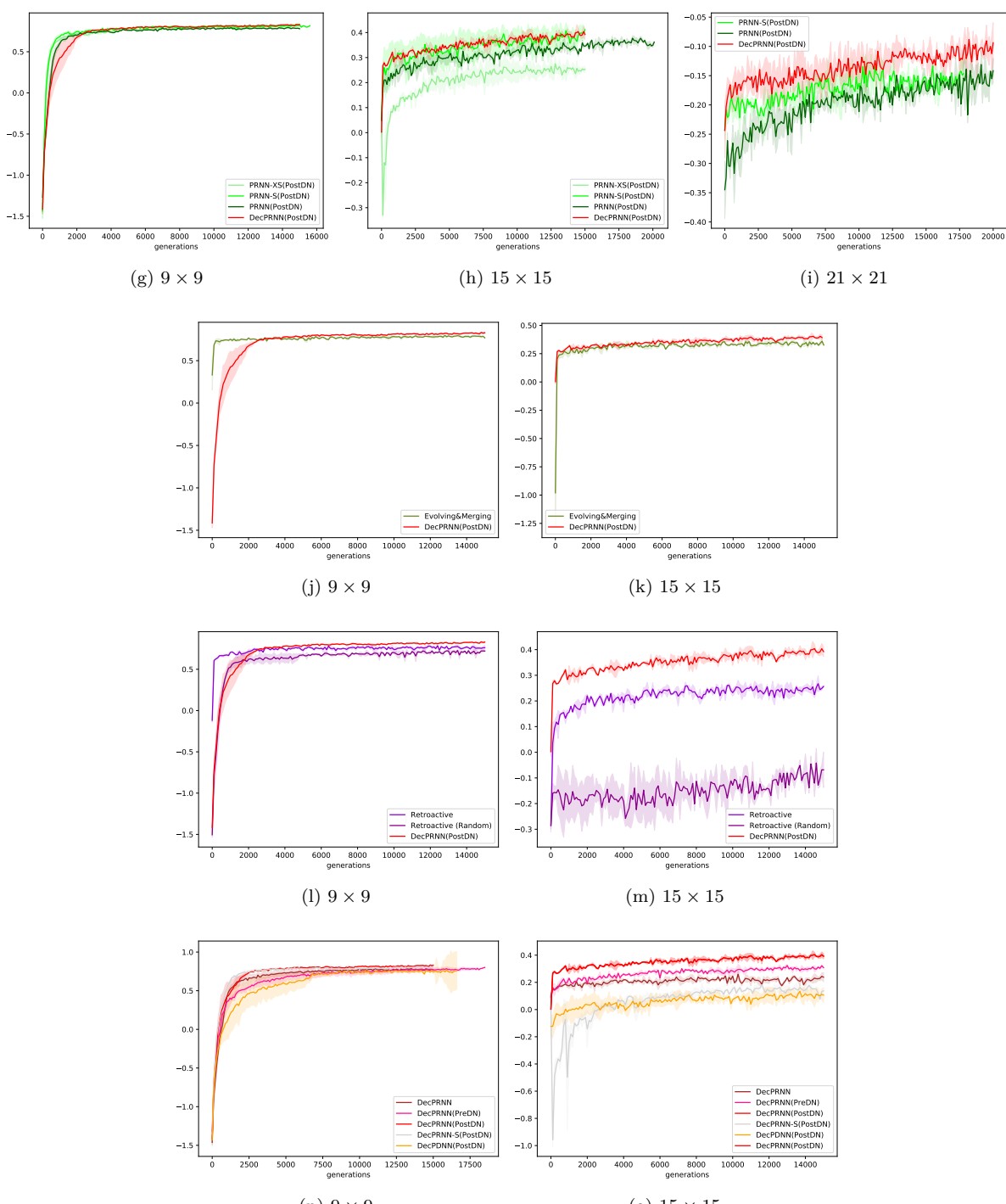

Figure 9: Plotting the mean and variance of the fitness score in validating tasks.

We train the learners on $9 \times 9$ mazes first and apply a warm start in $15 \times 15$, and $21 \times 21$ sequentially to reduce the meta-training cost. Also, it is not fair to compare DecPRNN(PostDN) with Evolving&Merging directly since we must start from a well-trained PRNN(PostDN) model in Evolving&Merging. Also, we occasionally reset the covariance to avoid the local optimum. Generally, we found that DecPRNN(PostDN) is not only one of the bests in performances but also converges fastly thanks to fewer meta-parameters. Model-based

learners such as Meta-LSTM are generally harder to reach convergence. We sometimes run models longer than the other models in case the models have not converged.

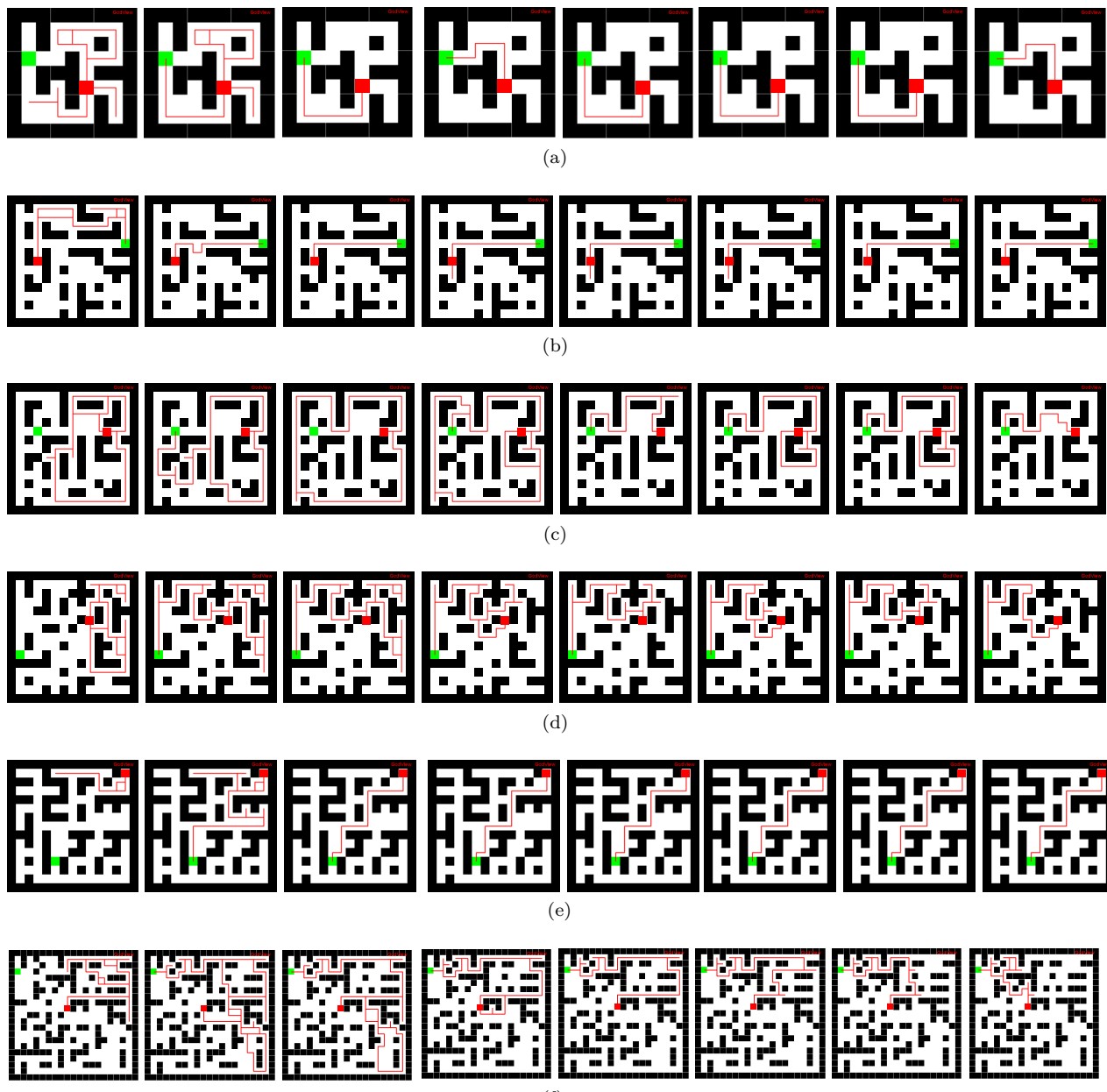

Figure 10: Example trajectories of the DecPRNN (PostDN) agents in each of the 8 rollouts in different mazes ($9 \times 9$, $15 \times 15$, and $21 \times 21$). The red squares mark the start position, the green square marks the goal, and the red lines denote the agents' trajectories.

### A.3.3 Lengths of the Life Cycles

The life cycle of the agents lasts for eight episodes. Each episode has at most 200 steps. The agent takes 1600 steps at most, but it is shortened as the agent performs better. The steps of a life cycle vary for different learners and different meta-training stages. In Figure 11, we show the changes in the average life cycle length for DecPRNN (PostDN) learners versus evolved generations, as well as the distribution of the life cycle length.

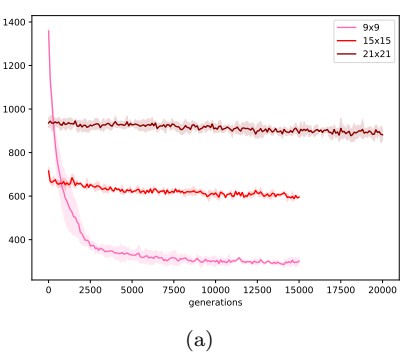
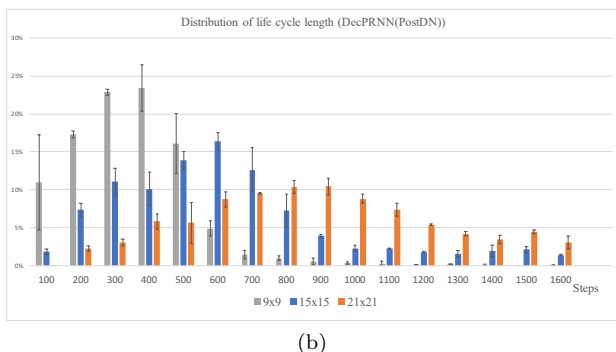

(a)                                                  (b)

Figure 11: Lengths of the life cycles of DecPRNN(PostDN) agent in $\mathcal{T}_{valid}$. (a) Changes in the average length in meta-training. (b) Distribution of the lengths at convergence.

### A.3.4   Case Demonstration[2]

We randomly sample several mazes and show the actual trajectory of the agents of each rollout in Figure 10. We demonstrate cases where the agents find or do not find the global optimum. This could give an intuitive interpretation of inner-loop learning. We can also observe the behaviors of *exploitation* to maintain high performance in the current rollout and *exploration* to reveal better routes for the following rollouts. For instance, the agent tends to explore new directions in case its previous rollout is not successful enough and takes the shortcut discovered in the previous rollout.

---

[2]Visualization of DecPRNN(PostDN) can be found at `https://www.youtube.com/watch?v=8EA8KqlCRzY`

