# OpenReview forum: "Evolving Decomposed Plasticity Rules for Information-Bottlenecked Meta-Learning"
_TMLR — Accepted by TMLR_

### Review · Reviewer_FY1B · 2022-06-29

**Summary Of Contributions:**

This paper introduces a method of learning plasticity rules in which the parameters for plasticity are learned at the neuron level instead of at the connection level. These plasticity rules are then used to update the agent’s NN weights from scratch (random initialization) during its lifetime. This method reduces the number of parameters to optimize from \Theta(n^2) to \Theta(n), where n is the number neurons. This decomposition of rules from connections to neurons should be of general interest to the plasticity community.

The motivation is that, in biology, the information (genome) used to initialize the organism is orders-of-magnitude less than the information (experiences/memories) accumulated over the organism’s lifetime; whereas, in SOTA plasticity-based (and other) meta-learning models in ML, the number of initially-learned parameters is at least as many as the number of parameters updated during the agent’s lifetime.

The proposed system is a direct, elegant method for addressing this mismatch: the “genome” is \Theta(n), and the “memories” are \Theta(n^2), so it satisfies the property of “genomics bottleneck”. In experiments in a random maze environment, the paper shows that the proposed method can perform as well as methods that do not satisfy the bottleneck, and may do even better when generalizing to OOD tasks. Visualizations show that the plasticity operates in a smoother way than prior methods.

**Broader Impact Concerns:**

I could not see any broader impact concerns.

**Requested Changes:**

### Changes critical for acceptance

- Abstract: Reduce generalization claim, e.g., “…and deteriorate the generalization capacity” -> “and potentially deteriorate the generalization capacity”

- Introduction:
	- Clarify meaning of “innate ability/capability” or rewrite this point in a different way. It is a confusing way to start the paper because “innate capability” in the first sentence refers to trained ANNs, but later in the paragraph it refers to untrained BNNs.
	- More generally, the genome metaphor should be made immediately clear in the introduction, since the whole paper depends on it. E.g., currently, the fact that meta-parameters can be thought of as “genomes”, and NN weights as “memories”, is not made explicit until half-way through Section 3.
	- “(Miconi et al., 2019b;a)” -> “(Miconi et al., 2018;2019)”. The 2019 paper is duplicated in the references. The first one should be removed.

- Outer-Loop Evolution:
	- Describe differences between seq-CMA-ES and standard ES to make more clear why seq-CMA-ES is used. This is critical information for future researchers.

- Experiment Settings:
	- Make clear why $w_z = 0.8^{7-z}$ for $z > 1$ in the Long Life Cycle. Otherwise, it looks like this was set simply to favor the proposed method, which does best towards the end of the life-cycle.

- The maze domain must be more clearly motivated. E.g., answer the questions: What are other common domains for plasticity methods? Why is this maze domain chosen instead of those? Is a high level of memory needed for maze-solving for some reason?

- A more interpretable metric should be added, e.g., to Table 1. I suggest using % mazes solved. Without this, since this is a new benchmark, it is difficult for the reader to interpret the significance of the performance differences.

- Make clear what the error bars mean in Table 1 and in the Figures. It appears these are computed across mazes, not across independent trials of each algorithm. If there is indeed only one run of ES per method, I highly suggest increasing the number of runs since the only evaluation is in a toy domain with a relatively small number of weights. Even 3 runs per method should be sufficient to see if ES training is reliable, i.e., doesn’t depend too much on the initial centroid.

- Learning curves, like the ones in Figure 7 (in the Appendix) should be shown for all methods. Otherwise, the reader does not know whether the differences in meta-test performance come from convergence in meta-training or fundamental differences in the final models, as claimed. To match the story in the paper, we would expect to see all the curves eventually flatten-out in these figures.

- Figure 6 (or some version of it), along with some of the insights from A.1 should be moved to the main paper. It is the most interesting analysis in the paper, showing that the proposed method yields smooth learning behavior over the agent’s lifetime that is qualitatively different than other methods. Without this, the reader may think the difference is due to the reduction in parameters, which may be inherently preferred in a toy domain like mazes to avoid overfitting. So, Figure 6 should be moved to the main paper and/or additional comparisons to PRNN and Meta-LSTM should be performed with number of neurons chosen so that N_gene approximately equals N_gene of DecPRNN. Doing both would strengthen the paper further.

- 4.2.2
	- Make clear why the “selected models” were selected. Why not show all of the models?
	- The point “that generalizability to OOD environments is more related to the genomics bottleneck than Hebbian plasticity” appears too strong without experiments showing that simply reducing the number of hidden neurons in PRNN would not lead to similar performance as DecPRNN. Either these experiments should be performed, or this claim should be reduced, e.g., “is” -> “could be”.

- 4.3
	- Make clear why the experiments with long life cycles are so limited compared to those with short. E.g., was it just due to computational cost? Long life cycles are more interesting to the reader, so they should know any pitfalls that prevented further experiments with long life cycles. Because DecPRNN seems to have more advantage in longer life cycles, the paper would likely have a bigger impact if these experiments were more complete.



### Further changes to strengthen the work

- Algorithms:
	- Problem Settings:
		- This paragraph defining the standard meta-learning setup is difficult to read through. Consider reorganizing, e.g., defining “meta-training” before “meta-training-training”. This paragraph might also be clarified with a figure.
	- Decomposed plasticity:
		- “It is orders…” -> “For large $n_x$ and $n_y$, it is orders…”
		- Make clear why this decomposition method is chosen instead of some other $O(n)$ method, e.g., learning an $O(n)$-sized function of the $v$’s instead of simply taking their outer-product. Is it just because outer-product is the simplest approach that works? Is it somehow sufficient for capturing complex learning rules? Are there any theoretical claims that can be made here?
	- Outer-loop evolution:
		- Make clear why ES is preferred here instead of gradient descent as in Miconi, et al.

- Experiment Settings:
	- Say why the final layer is non-plastic, i.e., is there some pitfall if they are plastic?

- For completeness, it could be useful to mention existing approaches that use hyper networks to generate both network weights and plasticity rules, since they may satisfy the genomics bottleneck condition, though they may be challenging to directly compare to because of the varying number of parameters depending on evolutionary dynamics (Risi & Stanley: “Indirectly encoding neural plasticity as a pattern of local rules” 2010; “A unified approach to evolving plasticity and neural geometry”, 2012).

- The curves for DecPRNN(PostDN) in Figure 6 are so simple that one might be able to characterize qualitatively how the weights are changing over time. A solid explanation for this extremely-simple-looking behavior would make the paper stronger.

- For Figure 6, make clear whether or not this visualization method has been previously used for visualizing meta-learning methods.
- Evolve&Merge should be added to Figure 6 since it is the most similar to DecPRNN in terms of n_gene and n_mem.

- The paper should explain why some comparisons take so long to converge with ES.

- Running the experiments in an established benchmarks could be the single biggest way to strengthen the paper.



### Additional clarity suggestions

- abstract:
	- “with verbose meta-parameters” -> “methods that have many more meta-parameters.”
	- $O(n^2)$ -> $\Theta(n^2)$
- introduction:
	- “BNNs Genomic Bottleneck” -> “BNNs: Genomic Bottleneck”
	- “such like human babes” -> “as in the case of humans” or remove completely
	- “few-shots” -> “few-shot”
	- Cite prior work for LBI
	- “neuron type” -> “neuron” (2x)
	- “from scratch” -> “from scratch, i.e., from random initialization,”
- figure 1:
	- “genomes that” -> “genomes, which”
	- “environment, which is” -> “environment, and are”
	- “massive information” -> “massive amounts of information”
- Modulated Plastic RNN:
	- Mention/cite previous work that uses modulated plastic RNNs
	- “input neurons respectively” -> “input neurons”
	- “actually use seq-CMA-ES” -> “use the ES variant seq-CMA-ES”
- Figure 2:
	- “crowed” -> “crowded”
- Standard Meta Testing:
	- Make clear how many hidden neurons are used in your approach.
	- “equal number of genomes” -> “approximately equal genome size”


**Strengths And Weaknesses:**

Strengths:
- The motivation makes sense: Due to their high number of parameters, existing plasticity methods are lacking in biological plausibility; the proposed method directly tackles this issue.
- The proposed method is an elegant attempt to addressing this issue head-on.
- Nice point that standard recurrent and self-attention models have inherent scalability issues as memory size increases.
- Nice point that information in memory should be larger than information in functionality
- Experimentally compares to many recent plasticity methods
- Compelling visualizations (although in the Appendix).
- The fact that the agent starts with random initialization in its lifetime is intriguing.
- Experimental results support the hypothesis that plastic networks that satisfy the genomic bottleneck and start from scratch do worse than comparisons at the beginning of the agent’s lifetime, but can do better by the end.

Weaknesses:
- The paper is fairly difficult to read. This is partly due to use of uncommon terms and terms that are not clearly defined early on in the paper. I have listed some possible ways to clarify in Requested Changes.
- The experimental random mazes environment is not clearly motivated.
- The experimental evaluation metric is not easily interpretable, so it is difficult to understand what exactly is being learned.
- Lack of comparison on existing meta-learning benchmarks, e.g., ones used by earlier plasticity approaches.
- It is not clear that all methods have converged.
- Important visualizations are in Appendix, not main paper.

---

> ### Author Response · Authors · 2022-07-21
> **Reply To Reviewer FY1B**
>
> We thank the reviewer for the thorough and detailed reviews and suggestions. As we agree with most of the suggestions and we promise to make revisions according to those suggestions in our next version, we respond to only part of the reviewer’s questions. Some of those questions are also raised by other reviewers, we post a reply to all reviewers regarding the commonly raised issues (including $w_z$, choosing the maze domain), and we will not repeat them in this part.
>
> - Describe differences between seq-CMA-ES and standard ES to make more clear why seq-CMA-ES is used
>
> We will add the following paragraph to the paper:
> For high-dimensional meta-parameter optimization, selecting hyper-parameters such as the covariance matrix is not trivial. It could end up in inefficient optimization and local optimums. To this end, CMA-ES has greatly improved computational efficiency by adapting covariance matrix using evolution path \cite{hansen2001completely}. However, it is at a price of increasing the per-step computational complexity from $\Theta(N_{\text{Gene}})$ to $\Theta(N_{\text{Gene}}^2)$, which is infeasible for large-scale ANNs. In seq-CMA-ES the covariance matrix is degenerated to $\Theta(N_{\text{Gene}})$ by preserving the diagonal elements only, which is both affordable and empirically more efficient compared with ES.
>
> - A more interpretable metric should be added e.g., to Table 1. I suggest using \% mazes solved.
>
> We think it is a good suggestion, and we promise to add this in our next version.
>
> - Make clear what the error bars mean in Table 1 and in the Figures
> - Learning curves, like the ones in Figure 7 (in the Appendix) should be shown for all methods
> - Figure 6 (or some version of it), along with some of the insights from A.1 should be moved to the main paper.
>
> The error bars are calculated from the collection of validating / testing tasks instead of running meta-training independently multiple times. The reason is related to computation costs. We promise that we will add at least 2 runs for each group in our next version. We also agree with the comments on Figure 6 and Figure 7, we will update it in the next version.
>
> - Make clear why the “selected models” were selected. Why not show all of the models?
>
> We tried to show the performances of all compared methods in one figure and found it messy. That is why we omit some of the methods that we think may not be so important for comparison in detail.  We will re-organize the experiments by comparing different methods in different groups.
>
> - Make clear why the experiments with long life cycles are so limited compared to those with short.
>
> We agree that experiments with long life cycles are more important than those with short ones, but we try to reduce the comparison within more expensive long life cycles. We do think that doing experiments in the short life cycle and long life cycle separately lead to confusion. We will try to replenish the long-life-cycle experiments of the other methods in our next version.
>
> - Make clear why ES is preferred here instead of gradient descent as in Miconi, et al.
>
> In section 5.2, we have some discussions on why we choose ES beyond Gradient for the outer-loop optimizer. And we also think that ES is potentially better than Gradient in agents fitting genomics bottlenecks (Considering Sparse Rewards, Long Life Cycle, Large Memory and Small-Size Meta-Parameters)
>
> - Say why the final layer is non-plastic, i.e., is there some pitfall if they are plastic?
>
> First, if we turn the output layer into a plastic layer, we add $\Theta(4n + 4m)$ meta-parameters while exempting $\Theta(nm)$ parameters. Since the last layer is relatively small, it would be unnecessary from the view of the genomics bottleneck. Second, we have also tried to turn it into a plastic layer, and there are indeed pitfalls. Similar situations happen to the modulation neurons. Thus, the modulation layer and the output layer are both non-plastic.
>
> - make clear whether or not this visualization method has been previously used for visualizing meta-learning methods.
>
> We think it is the first work to show the updating parameters in a compressed space continuously. Previous work (e.g., Najarro et al. 2020) have also shown a visualization of the inner loops, but not in a compressed space.
>
> - For completeness mention existing approaches that use hyper networks to generate both network weights and plasticity rules
>
> Thanks for the suggestion. We think it is very necessary to cite HyperNEAT and RelatedWorks and we will add them in the next version. HyperNEAT assumes that different neurons follow the same rule decided by hyper-networks, which shares much in common with Plasticity + Genomics Bottleneck. However, they are very computational costly under the mainstream computation libraries, which makes it very hard to scale up, and unacceptable in our cases. Compared with HyperNEAT, Decomposed Plasticity can be efficiently calculated using tensor operations.

---

> > ### Comment · Reviewer_FY1B · 2022-07-24
> > **follow-up question**
> >
> > Thanks for the responses. Seeing the full plots that you are planning to add should give a more complete picture of how your method fits in among existing methods.
> >
> > What did you think of the idea that "additional comparisons to PRNN and Meta-LSTM should be performed with number of neurons chosen so that N_gene approximately equals N_gene of DecPRNN"? Is it possible that having a lower number of parameters may inherently make learning easier in this experimental setup, or is there a clear reason why this wouldn't be the case?

---

> > > ### Author Response · Authors · 2022-07-28
> > > **Reply to the follow-up question of Reviewer FY1B**
> > >
> > > Thanks for reminding.  We forgot to mention that we will add comparisons of Meta-LSTM and PRNN with similar size of "genomes" (Meta-Parameters) as DecPRNN in our next version. We argued that fewer meta-parameters are beneficial for generalizing to Out-Of-Distribution tasks. Also, fewer meta-parameters do make the outer loop easier and the computational cost lower. However, the reduction of meta-parameters cannot compromise the computation capacity or the "memories" in the inner loop, or else there should be a pitfall in performance (as mentioned in section 5.3).

---

> > > > ### Comment · Reviewer_FY1B · 2022-08-25
> > > > **Thanks for the updates + follow-up questions/comments**
> > > >
> > > > Thanks for the updates; they definitely strengthen the work. Here are some relatively minor comments/questions/suggestions for the updated versions:
> > > >
> > > > Comments:
> > > > - the abstract is clearer and claims are valid
> > > > - need some citation at first mention of “dopamine neurons”, or say that they are introduced in this paper
> > > > - It’s funny that in the section on evolution strategies, the initial method is from 2017 and the improvements are from 2001 and 2008. Since the reference from 2017 is just one instance of evolution strategies, maybe rephrase with something like: "By following Evolution Strategies (ES) (Salimans et al., 2017a) the genotype shall be updated by" -> "The genotype can be updated using an evolution strategies (ES) approach, e.g., (Salimans et al., 2017a):"
> > > > - There are two Figure 4’s. Please try to make the text larger on both of those figures. Caption of second Figure 4 should say what the colors of the bars mean, if they mean anything. The second Figure 4 should also all be on the same page or split up into separate figures.
> > > > - Thanks for adding the failure rate metric. The improvements are now clearly substantial, especially on the larger mazes.
> > > > - Thanks for bringing Figure 6 to the main paper. The updated description is also intriguing.
> > > > - Should note in Figure 6 which methods are yours. This would be helpful to include in the legends of other figures as well.
> > > > - Thanks for doing additional trials on smaller Meta-LSTMs. These results make me more confident in your approach.
> > > >
> > > > Questions:
> > > > - In the introduction: what is meant by “forward-only”?
> > > > - Can you clarify what is meant by “sweet spot”? Is this a region where the task has been solved?
> > > > - “We believe building diverse and close-to-reality simulators is essential for the future development of AI. Recently much effort has been put into this topic (Chevalier-Boisvert et al., 2018; Yu et al., 2018), but there is still a long way to go.” Can you clarify how this is relevant to your approach?
> > > >
> > > > Suggested edits for clarity:
> > > > - variant -> various
> > > > - panorama -> full score
> > > > - devastating -> reducing
> > > > - Genomic Bottleneck -> The Genomic Bottleneck
> > > > - following aspects. -> following aspects:
> > > > - proposed decomposed plasitcity -> propose decomposed plasticity
> > > > - n is the hidden size of neurons -> n is the number of hidden neurons
> > > > - But we take a step further -> But, in order to satisfy the genomic bottleneck, we go a step further
> > > > - generate a nearly infinite number of different tasks -> generate endless new distinct tasks
> > > > - latterly -> later
> > > > - “Pitfalls are encountered if we further replace the output layer with plastic neural connections.” -> “We found that performance was lower if the output layer was plastic, and since it contains relatively few parameters, keeping it static does not violate the Genomic Bottleneck.”
> > > > - grids -> locations

---

> > > > > ### Comment · Reviewer_FY1B · 2022-08-31
> > > > > **Clarification of previous comments**
> > > > >
> > > > > Just wanted to note that my most recent suggestions for the authors would not affect my preference for acceptance, but would improve the clarity of the paper. The motivation is strong, the technical contributions are solid, but the presentation still could be significantly refined to improve how the paper is received by its intended audience.

---

> > > > > ### Author Response · Authors · 2022-09-01
> > > > > **Paper updated based on the suggestions**
> > > > >
> > > > > We thank the reviewer again for the constructive suggestions and very detailed reviews. We have updated our paper based on those suggestions.
> > > > >
> > > > > - need some citation at first mention of "dopamine neurons"
> > > > >
> > > > > Dopamine neurons are, of course, not first introduced in this paper. We add explanations and references to dopamine neurons in the updated version. Also, we revise the statements that is not correct enough. E.g. dopamine neuron is only one kind of modulator neuron.
> > > > >
> > > > > - There are two Figure 4's.
> > > > >
> > > > > We've fixed this problem and put figure 5 on one page.
> > > > >
> > > > > - Should note in Figure 6 which methods are yours.
> > > > >
> > > > > Actually, the first two figures in figure 7 (previous figure 6) are from our proposed methods. Each trajectory refers to a specific life cycle for certain tasks. We add additional clarifications in the caption of Figure 7 to avoid confusion.
> > > > >
> > > > > - In the introduction: what is meant by "forward-only"?
> > > > >
> > > > > There is indeed a misuse of "forward-only"; thanks for pointing it out. We want to emphasize here that the model is capable of learning by itself, without human intervention. We've rephrased it with "automated learning".
> > > > >
> > > > > - Can you clarify what is meant by "sweet spot"
> > > > >
> > > > > Yes, the sweet spot refers to the areas where the connection weights are at convergence and change less. We may also replace it with "optimum", but it is an observation from the trace demonstrations and not fully validated. We've added some explanations to this.
> > > > >
> > > > > - Recently, much effort has been put into this topic - Can you clarify how this is relevant to your approach?
> > > > >
> > > > > Here we want to mention works building intelligence on top of a simulator instead of a static dataset. We tried to explain it more clearly in the revised version.
> > > > >
> > > > > For the phrases and words mentioned, we've updated them in the revised paper. We sincerely appreciate the very detailed review.

---

> > > > > > ### Comment · Reviewer_FY1B · 2022-09-01
> > > > > > **A couple typos**
> > > > > >
> > > > > > Thanks for updating. Noticed a couple more typos when scanning the updates:
> > > > > >
> > > > > > - "full score" -> "full scope"
> > > > > > - "least learning rules" -> "fewest learning rules"

---

> > > > > > > ### Author Response · Authors · 2022-09-02
> > > > > > > **New version updated**
> > > > > > >
> > > > > > > Thanks for reminding. We've fixed some typos and citation issues and re-updated the paper.

---

### Review · Reviewer_Z376 · 2022-07-01

**Summary Of Contributions:**

The paper is situated in the field of meta-learning (Hebbian-like) plasticity rules for neural networks. A core challenge for meta-learning is the that number of (meta-)parameters scales quadratically with the number of neurons in the network (because there are 3-4 meta-parameters per connection and dense neuron connectivity is assumed). This is neither biological plausible (due to the genomic bottleneck for meta-parameters) nor suitable for data/compute-efficient meta-learning. To address this, the paper proposes a "neuron-dependent decomposition of those meta-parameters" (essentially a rank-1 matrix factorization), such that meta-parameters are learned per neuron and not per connection. This reduces the meta-parameters to be proportional to the number of neurons (and not to the square of the number of neurons). The approach is validated MetaMaze2D benchmark and compared to various baselines.

**Broader Impact Concerns:**

I don't see additional broader impact concerns besides the (potentially) increased carbon footprint caused by the additional computational requirements for meta-learning that are discussed in Section 5.2.

**Requested Changes:**


In certain parts, the paper makes claims that I consider too speculative and not connected to the scope of the paper and the empirical findings:
 - "Interestingly, human beings also experience similar cognitive decline when getting old; a reasonable guess is the explosive growth of human life cycle length in recent years goes far beyond our average life cycle length in evolutionary history." (page 9-10, this statement would need to be connected to research in biology)
 - "We are looking forward to simulating inner loops as long as the human life cycle, genomes (NGene) and memories (NMem) of human scale, which has the potential of revealing human-like general intelligence." (page 11, please avoid speculations about human-like general intelligence in a context where experiments have been done in a Maze world)

I would kindly ask the the authors to remove these claims (or rephrase them such that they are backed by evidence in the paper or by references).

Source code is not accessible for the reviewers - if feasible, I would ask the authors to make it accessible in a anonymous form for us.

**Strengths And Weaknesses:**

Strengths of the paper:
 * Clarity and readibility of the paper are on a good level
 * The proposed "decomposed plasticity" is well-motivated and a simple change that can be easily reimplemented
 * The experimental section contains a comparison to a relatively large number of baselines
 * Experiments contain a diverse set of settings in MetaMaze2D, including different maze sizes, different life cycles, different spaciousness/crowdedness, and OOD meta-testing
 * Illustrations of trajectories in Figure 5 are helpful to understand the meta-learned learning process of the agent over its life time

Weaknesses:
 * The proposed "decomposed plasticity" is a very special solution for reducing the number of meta-parameters. Alternatives such as sparse neuron connectivity or other low-rank matrix decompositions are not explored but could help to establish a more complete picture of "Information-Bottlenecked Meta-Learning"
 * Validation is limited to Reinforcement Learning in the discrete MetaMaze2D domain. Further validation on other problems and domains would be required to establish that the findings of the paper hold more broadly.
 * Some of the claims of the paper are too speculative and not backed by evidence (see Requested Changes)
 * As discussed by the authors in Section 5.2. "Limitations", further scaling up in terms of life cycle length, hidden size, depth of layers, and maze scale is not demonstrated and the findings of the paper remain restricted to rather small-scale settings.
 * Table 1 is not very accessible since it contains a lot of information. Accessibility could be improved by breaking it into several smaller tables or figures. Moreover, standard error of mean would be a better quantity for providing error bars than variance.
 * Similar to the point above, the text block containing the discussion of Table 1 is not accessible. It should be structured by text formatting or subparagraphs.
 * The fitness function of the long life cycle is non-trivial [w_z = 0.8^(7-z)] - is such manual engineering of the fitness function necessary and if yes, why?

---

> ### Author Response · Authors · 2022-07-21
> **Reply To Reviewer Z376**
>
> We thank the reviewer for the constructive reviews and suggestions. We will carefully address the suggestions, including removing some unverified and too strong points. As we agree with most of the suggestions and we promise to make revisions according to those suggestions in our next version, we respond to only part of the reviewer’s questions. Some of those questions are also raised by other reviewers, we post a reply to all reviewers regarding the commonly raised issues (including $w_z$, choosing the maze domain), and we will not repeat them in this part.
>
>
> - The proposed "decomposed plasticity" is a very special solution for reducing the number of meta-parameters. Alternatives such as sparse neuron connectivity or other low-rank matrix decompositions are not explored but could help to establish a more complete picture of "Information-Bottlenecked Meta-Learning"
>
> The motivation of decomposed plasticity is to reduce the number of meta-parameters (genomes as we defined) while preserving the number of adjustable parameters (memories as we defined). Reducing the neuron connectivity will also lessen the memories and thus the learning potential of the agent. Also, we've connected our proposal with biological plausibility (discussed in 5.1), which can not be easily satisfied with other settings.
>
> - Table 1 is not very accessible since it contains a lot of information.
>
> We will re-organize our experimental sections and try not to show Table 1 in our next version. The agents with different $N_{Gene}$ and $N_{Mem}$ can be shown more clearly with points in 2-D space where the x-axis and y-axis correspond to $N_{Gene}$ and $N_{Mem}$ respectively. We will replace table 1 with more easy-to-understand figures in our next version. And we will also rephrase the discussions to make it more clear.
>
> - Source code is not accessible for the reviewers - if feasible, I would ask the authors to make it accessible in an anonymous form for us.
>
> An anonymous version of the code is available at: https://anonymous.4open.science/r/EvolvingPlasticANN-D347/README.md

---

> > ### Comment · Reviewer_Z376 · 2022-08-26
> > **Feedback to revised version**
> >
> > I would like to thank the authors for a major revision of the manuscript based on our feedback. Overall, I think the paper has been improved. A few comments on the revised version:
> >  - The presentation of the results has been improved significantly. One drawback: text in many figures is not legible at 100% zoom level (typical printout). Could the authors increase the font size?
> >  - The part on "Visualizing Adaptive Components" is interesting but based purely on illustrations in a t-SNE  embedding space. Could some quantitative numbers on "short-term vibrations" and "long-term migrations" be defined to complement the qualitative discussion based on the visualizations?
> >  - Figure 1: the illustration is overall helpful but the part with the static/dynamic brain is problematic because it is clearly wrong that static/dynamic parts in the brain are spatially separated
> >  - Figure 2: The is a nice figure that can be very helpful. Could the authors clarify what the different "update" arrows are doing (maybe by referring to Equations in the main text).
> >  - there are a few parts where further references should be provided. E.g. for the part on "agent to adapt to non-stationary tasks continually during its lifetime ".
> >  - some wording is misleading, e.g. "The performances of model-based and plasticity-based learners diverge": the performances do not diverge but show different trends/deviate over time. Also I would propose to say "performance improves" rather than "grows" (Section 4.2.2). Also formulations like "[...] pitfalls are encountered" (used twice)/"devastating" are too colloquial and should be rephrased. Moreover, formulations like "it seems like" and similar are used several times and overall result in the impression that there still are many speculative parts in the paper while the focus should be on the actual findings backed by solid evidence.
> >  - it is a bit strange to stretch Figure 4 over two pages. Is there a reason not to have it on one page?
> >  - Section 2.4: "manipulate a network" -> "adapt a network"
> >  - (Salimans et al., 2017a) should not be the primary reference for Evolution Strategies, which dates back to the Seventies (Ingo Rechenberg).
> >
> > Overall, I think the paper is making good progress but I would still recommend an additional revision focusing on making the text more concise, less speculative, with more references and careful wording.

---

> > > ### Author Response · Authors · 2022-09-01
> > > **Paper updated based on the suggestions**
> > >
> > > We thank the reviewer again for the constructive suggestions. Based on those suggestions, we have updated our paper. Below we briefly mention those modifications.
> > >
> > > -  text in many figures is not legible at 100% zoom level
> > >
> > > Thanks for pointing it out; we increased the font size in the new version.
> > >
> > > - Could some quantitative numbers on "short-term vibrations" and "long-term migrations" be defined to complement the qualitative discussion based on the visualizations?
> > >
> > > Thanks for the suggestion; we added two measures to quantify the short-term and long-term behaviors. See Figure~8. We show the short-term vibrations and long-term migrations against the inner loop steps. They validate our proposal that plastic connection weights are dominated by long-term migrations, and hidden states are dominated by short-term vibrations.
> > >
> > > - Figure 1: it is clearly wrong that static/dynamic parts in the brain are spatially separated
> > >
> > > We remove the color separation of static/dynamic parts.
> > >
> > > - Figure 2:  Could the authors clarify what the different "update" arrows are doing
> > >
> > > We've explained different updates by referring to the equations in our revised version.
> > >
> > > - missing references.
> > >
> > > We added the reference in the updated version.
> > >
> > > - words & phrases
> > >
> > > We rephrase those words mentioned in our updated version.

---

> > > > ### Comment · Reviewer_Z376 · 2022-09-02
> > > > **Response from reviewer**
> > > >
> > > > I would like to thank the authors for taking my feedback into account and revising the manuscript! I don't have further comments.

---

### Review · Reviewer_hAdL · 2022-07-18

**Summary Of Contributions:**

The authors investigate the effects of imposing low-rank structure on differentiable Hebbian plasticity learning rules. These ideas are tested in a meta-learning / reinforcement learning setting. The method (DecPRNN) seems to perform reasonably well on a simple Maze navigation task, although other methods seem comparable (maybe slightly worse) to my eye.

**Requested Changes:**

The "Related Works" section should be re-written to make the contributions of this paper more clear. Please remove statements to the effect of "...plasticity and model-based learning... closely simulates the learning of BNNs" (see section 2.1). This assertion lacks justification as currently written. The authors should keep in mind that our current understanding of learning in BNNs is very limited.

I only understood the methods and motivation for this project after reading Miconi et al (2019, ICLR) in detail. For example, equation 2 in the submitted manuscript is not clear because the time index, t, is not incorporated into these expressions. I had to refer to Miconi to understand.

I found the discussion of "genomics bottleneck" to be confusing and unnecessary to motivate the paper. Similarly, I found the $\theta_\text{Gene}$ notation an unhelpful distraction / confusion. It seems like the ideas could be presented in a more straightforward manner.  If I understand correctly, the paper is suggesting a simple extension to prior methods that used differentiable Hebbian learning &mdash; the idea is to impose a low-rank structure on the meta-learning rule to reduce the number of trainable parameters to O(n + m) rather than O(n * m). The current way the paper is written obfuscates this because the intuition behind existing models is not explained, they are just listed in a citation dump.

Several sections and ideas seem to be superfluous. For example, sections 4.3 and 4.4 are very short and it is not clear what I should take away from them. Overall, I would like the see the paper be revised to key in on what the main takeaways are. It was honestly difficult for me to read and understand.

**Strengths And Weaknesses:**

Strengths:

* The authors show that simpler models (i.e. those with many fewer parameters) do reasonably well, if not better, on the Maze task studied here.

Weaknesses:

* I had a really hard time understanding the motivation behind this paper and the related works section (see below).
* The Maze task is not particularly challenging relative to some of the tasks studied in previous papers (e.g. Miconi et al 2019 apply this idea to Penn Tree Bank, language modeling task). Adding additional tasks and experiments would improve the paper.

---

> ### Author Response · Authors · 2022-07-21
> **Reply To Reviewer hAdL**
>
> We thank the reviewer for the constructive reviews and suggestions. We regret that our writings lead to some confusion. We will carefully address the suggestions, including removing some unverified and too strong points. Below we only try to clarify some of those problems. Some of those questions are also raised by other reviewers, we post a reply to all reviewers regarding the commonly raised issues, and we will not repeat them in this part.
>
> First, it is essential to clarify the difference between the work of Miconi et al. and ours. Both proposals address plasticity-based recursive neural structures. But, there are many differences between them, including:
> 1. different plasticity rules (decomposed plasticity versus eligibility traces)
> 2. different plastic layers (We apply plasticity to both input and recurrent layers, Miconi et al. apply it to recurrent layers only).
> 3. The models of Miconi et al. update the connection weights from meta-trained initial weights, while DecPRNN updates the connection weights from scratch (randomly initialized weights).
>
> We believe the 3rd difference is also very important because we believe effective plasticity rules should not necessarily restrict the start point of the connection weights but be effective whatever the initial weights are given. We have also shown that if we force the initial weights of the model in Miconi et al. to be random (namely Scratch-ET in our experiments), the performances suffer from pitfalls. It shows that the eligibility trace learning rules in Miconi et al. can not update connection weights from scratch (probably because the eligibility trace learning rules are too simple).
>
> Second, both the definition $\theta_{Gene}$ and $\theta_{Mem}$ are related to the essential contribution of this work. An intuitive explanation for our motivation is like this: Considering the vanilla RNN (or LSTM), the $\theta_{Mem}$ by our definition is at the scale of $\Theta(n)$, corresponding to hidden states; The $\theta_{Gene}$ by our definition is at the scale of $\Theta(n^2)$, corresponding to the connection weights. We ask this question: Do we really have to use $\Theta(n^2)$ parameters to drive only the $\Theta(n)$ memories? Or alternatively, can we use only $\Theta(n)$ parameters to drive $\Theta(n^2)$ memories? In other words, can we use much simpler learning rules to update a neural network without target function and back-propagation? What will it be like? And we recognize how similar this is to the genomics bottleneck. A $\Theta(n)$ parameter corresponds to the limited genomics ($\theta_{Gene}$), meaning lower inn-ability. However, A $\Theta(n^2)$ memory $\theta_{Mem}$ enables it to learn a lot by forward computation, meaning large memory and higher learning potential. The core idea of this paper is to use very few genomes (or what we call meta-parameters) to build strong BNN-like learning machines. We understand that there have been a lot of work in reducing parameters. But we are probably the first to raise the inspiring question and also propose an unusual idea, that we should reduce the parameters while increasing the memories.
>
> Going back to Miconi et al., as the initial connection weights ($W_0$) can not be arbitrary, they must be considered as part of $\theta_{Gene}$ (which is determined by genomes), thus making the genomes at least as large as the memories. We think it can be considered as an analogy to supposing that human babes are born with all the connections of their neurons "properly initialized by genomes," which can not be true due to the genomics bottleneck. To update a large neural network with fewer rules (fewer $\theta_{Gene}$), according to our proposed framework, is important to the generalization capability of the intelligence. And to this end, we add 4.2 to validate our point.

---

> > ### Comment · Reviewer_hAdL · 2022-07-25
> > **Brief feedback**
> >
> > I still think you should consider changing terminology "genomes -> meta-parameters" and "memories -> parameters". I personally think this would be easier for a reader to digest, although I won't argue for rejection on these superficial grounds. This is just a suggestion.
> >
> > The most concrete advice I can give is to re-write section 3 to be as clear and as self-contained as possible (e.g. I think the modulator variables $m_t$ defined in Eq 6+7 only show up in Eq 1? Reproducing equation 1 in section 3 could maybe help.). I am also not clear on how some of these baseline models were trained, e.g. could you clarify how "Meta-RNN" are trained relative to the approach used in [this paper (Wang et al. 2018)](https://doi.org/10.1038/s41593-018-0147-8), which would be a useful baseline to discuss and potentially include.

---

> > > ### Author Response · Authors · 2022-07-28
> > > **Reply to the feedback from Reviewer hAdL**
> > >
> > > We thank the reviewer again for the constructive suggestions. The notion of "Genomes" and "Memories" are "borrowed" from BNNs to clearly state our motivation. We didn't expect this to cause confusion in the following sections, especially when discussing algorithms. We initially use the notion "memories" instead of "parameters" to avoid confusion, because it might refer to both parameters and the hidden states. We agree that it might be more clear if we use the notion of "meta-parameters" instead of genomes. We will also use "adaptive parameters" to replace "memories" to avoid confusion. The changes will be included in the next version.
> > >
> > > We omitted some details in section 3, and we will make it more clear in our next version. We also notice that the mentioned paper "Prefrontal cortex as a meta-reinforcement learning system" (Wang et. al 2018) is closely related to ours, which should be added to the citation. The Meta-RNN or Meta-LSTM used in that paper is exactly the same as ours regarding inner-loop learning (check Algorithm 1 in our work, both take the triplet of (state, action, reward) as inputs). However, they differ in how they design and use the outer loop. The paper (Wang et. al 2018) follows mainstream meta-RL settings and employs a typical RL learner (e.g., actor-critic) to optimize the meta-parameters, while our work employs ES. In our opinion, the choice of the outer-loop optimizer is not crucial for differentiating different methods. It is also related to different motivations and interpretations of Meta-Learning. Wang et al. interpret Meta-RL as the combination of phasic dopamine and the prefrontal cortex. In contrast, we try to explain it as the combination of natural evolution and lifetime learning.

---

> > > > ### Comment · Reviewer_FY1B · 2022-08-01
> > > > **genomes and memories**
> > > >
> > > > I agree that it is tricky to read initially, but in the end, I think the "genome" and "memory" terms help to drive home the motivation of the paper. To me, the core contribution of the paper is: (1. Question) What happens if we take seriously the idea that meta-parameters correspond to genetic material? (2. Implication) Then, we need some kind of genomic bottleneck property, and (3. Implementation) a decomposition of plasticity parameters is a direct and effective way to achieve it.
> > > >
> > > > The genetic metaphor is important to make the contribution work, but it's still worth considering changing the terms to make it easier to understand. For example, "neural connections" could be more informative than "memories", and "genotype" could be more in line with the evolutionary ML literature than "genome".

---

> > > > > ### Author Response · Authors · 2022-08-05
> > > > > **Genotype & Phenotypes?**
> > > > >
> > > > > We thank the reviewer again for the suggestions. We think it is a good idea to use "genotype" instead of genomes. But for "memories," it still seems confused to describe using other notions. We try to classify different types of "phenotypes"(or "parameters") for the agents; there are four types:
> > > > >
> > > > > - Learning Functionalities (Learning Rules)
> > > > > - Static Neural Connections
> > > > > - Plastic Neural Connections
> > > > > - The recursive Neuronal States
> > > > >
> > > > > Originally, the first two in our paper are referred to as "genomes" or "meta-parameters"; The last two in our paper are named "memories ."To avoid all this confusion, we will avoid using the word "memories" and try to re-organize all those concepts as follows and clearly state them with figures or tables in our next version:
> > > > >
> > > > > - The "genotype", a.k.a "meta-parameters", can decide the initial "phenotype"
> > > > > - The "phenotype" is composed of static components (Learning Functionalities & Static Neural Connections) and adaptive components (Plastic Neural Connections & Recursive Neuronal States).
> > > > > - The static components are decided by "genotype" and kept static in the agent's life cycle
> > > > > - The adaptive components are started from scratch and updated continually in the agent's life cycle.
> > > > >
> > > > > We hope it will be clearer in this way.

---

### Author Response · Authors · 2022-07-21
**Addressing the Common Concerns of All Reviewers**

Since reviewers have common questions on some issues, we will first answer those issues. We have also noticed that the experiment section needs to be revised, and we are planning to supplement additional results to support our standpoints. We will describe how we plan to reorganize the experiment section in our next version. It would require an extension to the review deadline. We want to update our paper by no later than Aug. 20th. We hope that decisions can be made after that time.

### Addressing Common Concerns

- Why $w_z=0.8^{(7-z)}$ for $z>1$ in the Long Life Cycle

Notice that $w_z$ influences meta training only, which is similar to weights of loss functions in canonical supervised learning. Here, we want the evolution to avoid abandoning those individuals that perform poorly at the beginning of their life cycle but get better in the end, which means they have good learning capability. This would be more clear if we plot $w_z$ as a function of $z$ as follows. If we view it from the canonical Meta-learning settings, it is reasonable to set $w_z=[0, 0, 0, 0, 1, 1, 1, 1]$. The first four rollouts can be considered as "Support Set," and the last four rollouts are considered as "Query Set." This could mean that we encourage the agent to do all explorations for the first four rollouts and exploitations for the last four rollouts. But we do not want such "hard" settings. We want the agent to explore, learn, and do better gradually during its lifetime. Thus we are using a "soft" setting for $w_z$. We found that using a hard "Support Set" and "Query Set" setting is not good for convergence for all the compared methods in meta-training. The soft setting is better in most cases. But we do not exhaustively search for the best settings and it seems that the whole framework is relatively robust to this hyper-parameter. In other words, it might also be ok to use $0.6^{(7-z)}$ or $0.7^{(7-z)}$.

- Why Choosing Maze Domain?

We will clearly state why we choose the maze domain in our next version. There are a variety of environments for bench-marking meta-learning and sequence modeling. Generally, many of those are suitable for few-shot learning. For instance, we have explored variant mujoco environments (where the topologies are changed with tasks) such as ants and humanoids. However, we find that model-based and plasticity-based agents can adapt to various tasks within very few steps. A reasonable guess of this phenomenon is that the agent captures patterns of all the tasks they have met in meta-training. In our opinion, in such cases, the agents can adapt by simple "task pattern recognition" and "policy selection." It violates the principle that motivated our research, that most knowledge should be acquired by inner-loop learning instead of outer-loop learning.
In contrast, the random maze requires a more significant number of explorations and memories to reach a higher score. To simplify, we are more interested in domains requiring a long life cycle to do reasonably well. More complex tasks such as extra long text modeling are also suitable but would require much larger computation resources. Also, it would be hard to reach a valuable conclusion using only the simplest DecPRNN; we probably need to implement decomposed plasticity and recursion in more sophisticated structures, making validating our proposal less clear. We are considering solving long text modeling tasks in our future work.

### Plans to Revise the Experiment Sections

We try to thoroughly and fairly compare different plasticity and model-based methods (till now, probably the most integrated survey of plasticity-based methods), but this has indeed made Table. 1 lengthy and hard to follow. We also find it messy to plot the per-rollout performance of all the methods in one figure. We will try to group the methods for comparison in different figures to explain different points. E.g., we will put DecPRNN, DecPRNN(PreDN), and DecPRNN(PostDN) in one figure to illustrate the contribution of neural modulation.
Due to the limitations of the computation budget, we have experimented on both short life cycles and long life cycles. Indeed, as Reviewer FY1B stated, we focus on more on long life cycles. We are trying to add experiments in long life cycles. Also, it would be perfect to run meta-training independently for at least 3 runs. We will try our best to add one additional group of meta-training, and then plot the variances with respect to independent meta-training instead of the variances of different tasks in tests or validation.

---

### Decision · Action_Editors · 2022-09-09

**Recommendation:** Accept as is

**Comment:**

This paper focuses on a few contributions related to meta-learning agents, primarily the decomposed plasticity approach introduced by the authors to reduce the number of meta-parameters required for supporting meta-learning.  The authors also evaluate a form of structured artificial neuromodulation.

The reviewers largely understood the relevance of the problem.  For example, reviewer FY1B noted “Due to their high number of parameters, existing plasticity methods are lacking in biological plausibility; the proposed method directly tackles this issue.”  Reviewer Z376 noted “proposed ‘decomposed plasticity’ is well-motivated and a simple change that can be easily reimplemented”.  Some reviewers (FY1B and hAdL) found parts of the original presentation unclear.  Reviewer hAdL engaged with the authors concerning some of the terminology and notation, which the authors adjusted.

Based on the initial feedback from reviewers, the authors undertook substantial revisions.  Note that this paper lagged relative to the target timeline for TMLR submissions, primarily due to relatively substantial revisions being performed as part of the author-reviewer discussion phase.  The authors updated the paper, making the notation and formalism more self-contained, better motivating the choices in the work, and refining the claims made in light of the literature and the contributions made.  The authors also adjusted the contents of the main paper, and now include material in the main text that had previously been in the appendix.

After revisions, reviewers were largely satisfied with the updates. Reviewers FY1B and Z376 issued a few additional more minor pieces of feedback and the authors implemented additional changes. All reviewers ultimately either recommend acceptance or lean towards acceptance.

One shortcoming raised by all reviewers has to do with the exclusive reliance on the maze task.  Essentially, reviewers felt that inclusion of other tasks may increase confidence in the generality of the findings and that the maze is relatively simple.  The authors respond to this by further motivating the use of the maze, noting that the maze has partial observations, substantial procedural diversity, and a long life cycle.  They also note that the maze is compatible with limitations on computational resources. While this is helpful to clarify what about the maze made it suitable, I don’t think the features of the maze task identified by the authors are exclusive to it, so in principle other tasks with these features could also be considered.  However, application of the proposed method to additional tasks is somewhat onerous, and I think the evaluation is sufficient for a paper of this scope.

Very minor points:
- “Phenoype” is misspelled in figure 2.
- The observation in this work is a partial state feature (along with previous action and reward). The partial state feature is notated with s_t (“the 3 × 3 grids observed (s_t)” ) which could easily be mistaken for the whole state, so this is probably a suboptimal choice of notation.

---

> ### Author Response · Authors · 2022-09-14
> **Camera-ready version uploaded**
>
> We apologize for the typos and we fix the minor problems we found. To avoid confusion, we replace the state descriptor $s_t$ with the frequently used notation $o_t$ for partial observations. We have uploaded the camera-ready version.
>
> A visualization video for the proposed plastic neural network is released at https://www.youtube.com/watch?v=8EA8KqlCRzY
>
> We sincerely thank all the reviewers and the action editor for the constructive feedback and suggestions, which is helpful and crucial for improving our work.